# Measurement of formic acid, acetic acid and hydroxyacetaldehyde, hydrogen peroxide, and methyl peroxide in air by chemical ionization mass spectrometry: airborne method development

Victoria Treadaway[1], Brian G. Heikes[1], Ashley S. McNeill[1,2], Indira K.C. Silwal[3,4] and Daniel W. O'Sullivan[3]

[1]Graduate School of Oceanography, University of Rhode Island, Narragansett, RI 02882, USA
[2]Department of Chemistry, The University of Alabama, Tuscaloosa, AL 35401, USA
[3]Chemistry Department, United States Naval Academy, Annapolis, MD 21402, USA
[4]Forest Bioproducts Research Institute, University of Maine, Orono, ME 04469, USA

*Correspondence to*: Victoria Treadaway (vtreada@uri.edu)

**Abstract.** A chemical ionization mass spectrometry (CIMS) method utilizing a reagent gas mixture of $O_2$, $CO_2$, and $CH_3I$ in $N_2$ is described and optimized for quantitative gas-phase measurements of hydrogen peroxide ($H_2O_2$), methyl peroxide ($CH_3OOH$), formic acid (HCOOH), and the sum of acetic acid ($CH_3COOH$) and hydroxyacetaldehyde ($HOCH_2CHO$; also known as glycolaldehyde). The instrumentation and methodology were designed for airborne in situ field measurements. The CIMS quantification of formic acid, acetic acid, and hydroxyacetaldehyde used $I^-$ cluster formation to produce and detect the ion clusters $I^-(HCOOH)$, $I^-(CH_3COOH)$, and $I^-(HOCH_2CHO)$ respectively. The CIMS also produced and detected $I^-$ clusters with hydrogen peroxide and methyl peroxide, $I^-(H_2O_2)$ and $I^-(CH_3OOH)$, though the sensitivity was lower than with the $O_2^-(CO_2)$ and $O_2^-$ ion clusters, respectively. For that reason, while the $I^-$ peroxide clusters are presented, the focus is on the organic acids. Acetic acid and hydroxyacetaldehyde were found to yield equivalent CIMS responses. They are exact isobaric compounds and indistinguishable in the CIMS used. Consequently, their combined signal is referred to as "the acetic acid equivalent sum." Within the resolution of the quadrupole used in the CIMS (1 m/z), ethanol and 1- and 2-propanol were potential isobaric interferences to the measurement of formic acid and the acetic acid equivalent sum, respectively. The CIMS response to ethanol was 3.3% that of formic acid and the response to either 1- or 2-propanol was 1% of the acetic acid response; therefore, the alcohols were not considered to be significant interferences to formic acid or the acetic acid equivalent sum. The multi-reagent ion system was successfully deployed during the Front Range Air Pollution and Photochemistry Éxperiment (FRAPPÉ) in 2014. The combination of FRAPPÉ and laboratory calibrations allowed for the post-mission quantification of formic acid and the acetic acid equivalent sum observed during the Deep Convective Clouds and Chemistry Experiment in 2012.

**Keywords:**

Chemical ionization mass spectrometry, multi-reagent ion, formic acid, acetic acid, hydrogen peroxide, methyl peroxide, hydroxyacetaldehyde, cluster-ion chemistry

# 1 Introduction

Formic acid (HCOOH, hereafter referred to as HFo), acetic acid ($CH_3COOH$, hereafter referred to as HAc) and hydroxyacetaldehyde ($HOCH_2CHO$, commonly referred to as glycolaldehyde, and hereafter abbreviated as GA) are oxygenated volatile organic compounds (OVOCs) found in remote and urban environments in both gas and particle forms. Primary emissions for both acids include vegetation, agriculture, biomass burning and motor vehicle emissions (Khare et al., 1999; Paulot et al., 2011). Secondary sources also play a substantial role in the formation and distribution of HFo and HAc and include photochemical production from gaseous VOCs and OVOCs of biogenic and anthropogenic origin, biomass burning, and primary and secondary organic aerosols (Khare et al., 1999; Paulot et al., 2011). Both organic acids are photochemically long lived (>10 days with respect to oxidation by HO) and their removal is primarily by dry and wet deposition at the Earth's surface (Paulot et al., 2011). Away from the earth's surface, these acids represent a relatively long-lived intermediate product in the oxidation of organic matter. However, there is a scarcity of organic acid measurements in the upper troposphere with which to compare and assess photochemical and transport theory. Millet et al. (2015), Reiner et al. (1999), and Talbot et al. (1996) reported vertical profiles for HFo and HAc; however, only Reiner et al. and Talbot et al. sampled above 7 km. In remote environments, HFo and HAc are the primary acids establishing the pH of cloud water and precipitation (Galloway et al., 1982). HFo and HAc partitioning between gas and aqueous phases is pH dependent. In the aqueous phase, both HFo and HAc remain in the protonated form below their pKas of 3.75 and 4.76 (T=298.15 K), respectively (Johnson et al., 1996). As emission controls on anthropogenic $NO_x$ and $SO_2$ continue to decrease the contributions of these gases to precipitation acidity, the organic acids are expected to compose a larger fractional contribution to acidity in cloud water and precipitation.

Hydroxyacetaldehyde (or glycolaldehyde, GA) is formed by the HO oxidation of biogenic VOCs such as isoprene and methyl vinyl ketone (Lee et al., 1998; Tuazon and Atkinson, 1989) and by the HO oxidation of unsaturated anthropogenic VOCs like ethene (Niki et al., 1981). GA has also been measured in smoldering biomass burning plumes and can be up to 1% of the gaseous carbon detected in fire emissions (Johnson et al., 2013; Yokelson et al., 1997). Table S1 provides a summary of literature surface and aircraft measurements for GA in urban, biomass burning, biogenic, and mixed environments. GA's primary loss is by HO oxidation and wet deposition (Bacher et al., 2001). The effective Henry's Law constant for GA (70 M/hPa) is surprisingly large (Betterton and Hoffmann, 1988) and an order of magnitude larger than that for HAc (7.8 M/hPa) at a temperature of 288 K (Johnson et al., 1996; results below). GA is more likely than HAc to be removed by precipitation during transport through deep convection based upon model work by Barth et al. (2003) and Bela et al. (2016). Unpublished model results from Bela et al. showed a ten times greater removal of GA relative to HAc in a simulated DC3 deep convective storm.

There is a need to distinctly measure HAc and GA throughout the depth of the troposphere. They provide a test point for the processing of VOCs by different photochemical mechanisms. There are multiple precursors that, depending on the chemical mechanism, will lead to different portions of HAc and GA as second generation or later products. For example, while isoprene is an important precursor for GA it is thought to be insignificant for HAc (Lee et al., 1995b; Paulot et al., 2011). However, isoprene is also a significant source for peroxy acetyl radical, which reacts with $HO_2$ to form HAc (Khare et al., 1999; Paulot et al., 2011). In addition, GA is relevant to the tropospheric ozone budget (Lee et al., 1995b; Petitjean et al., 2010) and HAc directly effects precipitation acidity (Khare et al., 1999; Paulot et al., 2011). Finally, both GA and HAc are participants in the formation and growth of organic aerosols and in aerosol photochemical processing (Carlton et al., 2006; Fuzzi and Andreae, 2006; Lee et al., 2006; Perri et al., 2010; Yu, 2000). Airborne platforms provide one vehicle for instrumentation to measure these compounds throughout the depth of the troposphere (e.g., Le Breton et al., 2012; Lee et al., 1998; Millet et al., 2015; Talbot et al., 1996) and this adds an additional need for "fast" instruments, especially for situations in which spatial-temporal scales are relatively small, such as in the boundary layer or near convective clouds.

In recent years there has been an increase in the number of atmospheric gas-phase species measured using chemical ionization mass spectrometry (CIMS) (Huey, 2007). The major advantages of CIMS include rapid response times with high sensitivity and selectivity (e.g., Bertram et al., 2011; Crounse et al., 2006). Previous studies have successfully measured gas-phase HFo and HAc via negative-ion mode CIMS using trifluoromethoxy anion ($CF_3O^-$), iodide ($I^-$) or acetate ($CH_3COO^-$) as the reagent ion (e.g., Amelynck et al., 2000; Le Breton et al., 2012; Brophy and Farmer, 2015; Veres et al., 2008). Yuan et al. (2016) reported HFo and HAc using a $H_3O^+$ ToF-CIMS. Brophy and Farmer (2015) developed a dual reagent ion system with $I^-$ used for HAc and $CH_3COO^-$ used for HFo, and in which the reagent gases are added in an alternating sequence. However, to our knowledge, to date only one group has reported results for both HFo and HAc using an $I^-$ CIMS (Lee et al., 2014). Proton-transfer-reaction mass spectrometry has been used to quantify HFo and HAc with $H_3O^+$ as the reagent ion (e.g. Müller et al. 2014), although, Wisthaler reports the sum of HAc and GA (Armin Wisthaler, pers. comm. 2015) using this methodology.

O'Sullivan et al. (2018) and Heikes et al. (2017) described a CIMS instrument for the airborne measurement of peroxides called PCIMS. In the course of developing the PCIMS, the opportunity presented itself to investigate the sensitivity of HFo and HAc to multiple reagent ions, specifically $I^-$ and $O_2^-$. The PCIMS system was originally developed for the Deep Convective Clouds and Chemistry (DC3) experiment (O'Sullivan et al., 2018) and modified in post-mission calibration work. This modified system was then used in the Front Range Air Pollution and Photochemistry Éxperiment (FRAPPÉ) experiment (Treadaway, 2015) and modified again post-mission. The reason behind the post-mission DC3 development involves serendipity and foresight. Prior to the DC3 mission, the PCIMS underwent optimization for the measurement of hydrogen peroxide and methyl peroxide and, before settling on a $CO_2$-in-air reagent gas for the peroxides, a $CH_3I$ in $N_2$ reagent gas was tested (O'Sullivan et al., 2018). $I^-$, derived from $CH_3I$, proved to provide sufficient sensitivity for HP but not for MHP which was a critical species for the PCIMS, especially for the identification of deep convective storms during DC3 (O'Sullivan et al., 2018). $CH_3I$ is a "sticky" gas and, even though the reagent storage cylinder, regulator and transfer lines had been flushed, there remained a finite amount of $CH_3I$ in the system, which bled off the reagent line's interior surfaces. Evidence of this was observed at m/z ratios of 127 ($I^-$), 145 ($I^-(H_2O)$), and 147 ($I^-(H_2^{18}O)$). It was further noted that in addition to a m/z signal at 161 ($I^-(H_2O_2)$), there were m/z signals at 173 and 187, which were ascribed to HFo ($I^-(HFo)$) based on the work of Le Breton et al. (2012) and to HAc ($I^-(HAc)$), respectively. In DC3, the m/z signals at 173 and 187 were recorded with the expectation that post-mission laboratory calibration work would allow HFo and HAc to be quantified in the upper troposphere. This calibration work appeared to be successfully accomplished in the laboratory and validated in-flight during the FRAPPÉ mission in 2014. However post FRAPPÉ, GA, a potential isobaric interference, was confirmed for $I^-$ chemistry with a relative response of approximately 1:1 for HAc:GA. We necessarily report the m/z 187 signal as the "acetic acid equivalent sum" (AAES) of HAc and GA in our prior DC3 and FRAPPÉ datasets (data reporting in progress).

This study details the detection and quantification of HFo and AAES using a multi-reagent ion CIMS. The multi-reagent ion PCIMS is unique as it allows the detection of HFo and AAES, as well as, hydrogen peroxide ($H_2O_2$, hereafter referred to as HP) and methyl peroxide ($CH_3OOH$, hereafter referred to as MHP). The multi-reagent ion gas system blends a $CO_2$ in air mixture and a $CH_3I$ in $N_2$ mixture with pure $N_2$. This is different from other multi-reagent ion systems such as Brophy and Farmer (2015) as the two reagent gases are added simultaneously and tuned such that $I^-$, $O_2^-$, and $O_2^-(CO_2)$ ion cluster chemistries are operable. O'Sullivan et al. (2018) presented PCIMS measurements for HP and MHP using $O_2^-(CO_2)$ and $O_2^-$, respectively. Heikes et al. (2017) presented an ion-neutral chemical kinetic model to simulate the ion chemistry presented here and in O'Sullivan et al. Here, we report the results of the PCIMS calibration work with $CH_3I$ for HP, MHP, HFo, and HAc and interference work with ethanol, propanol and GA to determine: 1) the nominal $CH_3I$ concentration inadvertently used in DC3, 2) pressure and humidity dependent sensitivity factors for these analytes using $I^-$ cluster chemistry, 3) interference characterization

of a few common trace atmospheric gases, and 4) initial DC3 and FRAPPÉ HFo and AAES observations by the PCIMS instrument. The I⁻ molecule cluster kinetics were described in greater depth in Heikes et al. (2017).

## 2 Methods

### 2.1 Field Campaigns

The Deep Convective Clouds and Chemistry (DC3) field campaign was conducted in the central United States in May and June 2012. The PCIMS was on board the National Center for Atmospheric Research Gulfstream-V aircraft (HIAPER, UCAR, 2005), which flew 22 research flights ranging west to east from the Colorado Front Range to North Carolina, north to south from Nebraska to the Gulf of Mexico and from the boundary layer to 13 km. A description of the project, platforms, instrumentation, and measurements can be found in Barth et al. (2015).

The Front Range Air Pollution and Photochemistry Éxperiment (FRAPPÉ) consisted of 15 research flights in July and August 2014. The PCIMS was flown on the National Center for Atmospheric Research C-130 (UCAR, 1994) and primarily over the northern Colorado Front Range from the boundary layer to 8 km. FRAPPÉ was the first campaign using the two-syringe microfluidic calibration system (Sect. 2.4) and the three-mixture blended reagent ion scheme (Sect. 2.3). The project was a multi-agency, multi-investigator program and details of the experiment are available online
(https://www.eol.ucar.edu/field_projects/frappe and http://discover-aq.larc.nasa.gov/).

### 2.2 Instrumental Configuration

Continuous gas analysis was performed using a CIMS (THS Instruments, Inc., Atlanta, GA) in negative ion mode. The sample and analytical systems were based on the Slusher et al. (2004) design. Our modified CIMS, referred to as PCIMS, is depicted schematically in Fig. 1a, and instrumental settings are listed in Table 1. PCIMS was specifically designed to meet
engineering standards for use on HIAPER (O'Sullivan et al., 2018). Critical system elements include a gas sample delivery inlet with calibration system and the PCIMS, which is composed of a reagent gas blending system, ion generation and air sample reaction system, ion selection (declustering, ion guide, and quadrupole), multi-ion counting detector, and vacuum system.

Ambient or laboratory sample air entered the PCIMS system through a PFA Teflon® inlet and transfer line. In the laboratory, synthetic air mixtures were delivered to the inlet using PFA Teflon®. In airborne field work, a HIAPER Modular
Inlet (HIMIL) was hard mounted on the fuselage and extended beyond the aircraft boundary layer. The HIMIL is aerodynamically designed to minimize the collection, volatilization, and subsequent analysis of large aerosol and cloud drop/ice material as an artifact in gas measurements. The HIMIL and gas transfer lines were heated to 313 K in DC3 and 343 K in FRAPPÉ to minimize artifacts caused by the adsorption/release of the target gases onto/from inlet surfaces. The HIMIL inlet surfaces were lined with PFA Teflon® tubing. Field calibrations (Sect. 2.4) were performed by standard addition to the sample
air stream. The PCIMS responded linearly to the analyte gases measured at a fixed sample pressure and water vapor mixing ratio and species sensitivity was determined using a single calibration gas mixing ratio for each analyte. Analytical blanks (Sect. 2.5) were determined by passing the sample air stream, with or without calibration gas, through serial Carulite 200® and NaOH traps. As discussed below, PCIMS sensitivity varied with sample pressure and water vapor mixing ratio.

In PCIMS, the sample air passed through a series of chambers to form, select, and quantify the organic acid ion clusters.
The first chamber was the ion-sample reaction cell, RXN in Fig. 1a. In the reaction cell, the sample air was mixed with a reagent ion stream (Sect. 2.3) of which the bulk was pure nitrogen and controlled by mass flow controllers (MFCs). The total flow through the reaction cell was fixed at 4.68 slpm (standard liters per minute; T = 273.15 K and P = 1013.25 hPa) and the mean

transit time through the reaction cell was 17.8 ms . The reagent gas mixture was passed through a commercial electrostatic eliminator (model P2031-1000, NRD, Inc., Grand Island, NY), which initially contained 20 mCi of $^{210}$Po, an alpha emitter, and thus developed the requisite reagent ion stream (e.g., Heikes et al., 2017). The electrostatic eliminator was pre-treated with sodium bicarbonate per THS recommendation (THS Instruments, Inc., Atlanta, GA) to trap emitted residual nitric acid vapor present in the ion source from its manufacture. The RXN cell sample inlet and outlet critical orifices were of fixed diameter and optimized by THS to have a reaction cell pressure of 22 hPa, given the vacuum pumps and reagent gas system employed. This pressure was stated to provide the maximum yield of cluster ions and peak sensitivity and was not further evaluated, although the work of Iyer et al. (2016) suggested a higher RXN cell pressure could lead to higher sensitivities for analyte molecules with 8 or fewer atoms. For laboratory work in Narraganset, RI, and Annapolis, MD, the reagent nitrogen and the sample flow rates were effectively constant at 2.0 and 2.68 slpm, respectively. However, in airborne operations, the inlet pressure decreased with altitude, the sample flow decreased proportionately because of its fixed orifice area and the reagent $N_2$ flow was necessarily increased to maintain a constant RXN cell pressure. Note: a variable critical orifice sample inlet was unavailable at the time of DC3 and, while available for FRAPPÉ, was not flown then to best evaluate the DC3 post-mission calibrations and their use in DC3 to recover HFo and HAc in that program. Consequently, instrument response in this work varied with sample inlet pressure or sample flow rate and was quantified in the laboratory and during FRAPPÉ (Treadaway, 2015; Heikes et al., 2017).

**2.3 Reagent Gas**

The reagent gas during DC3 was $CO_2$ (400 ppm, 0.080 slpm) in ultrapure air blended with pure $N_2$ (Scott-Marrin, Riverside, CA). The $CO_2$ and air reagent gas flow rate was optimized for HP and MHP signal response (O'Sullivan et al., 2018). An iodide source gas (iodomethane, $CH_3I$), was used during pre-DC3 experiments as a potential reagent gas and was found to effectively cluster with HP but not with MHP (O'Sullivan et al., 2018). A residual amount of $CH_3I$ had adsorbed onto the reagent gas handling interior surfaces and was found to bleed off this plumbing in DC3. Post-DC3, a laboratory $CH_3I$ in ultrapure $N_2$ mixture was developed which reproduced the $I^-$ available during DC3. The $CH_3I$ reagent gas was prepared similarly to Le Breton et al. (2011) but without the addition of water. Liquid $CH_3I$ (Sigma-Aldrich, St. Louis, MO) was first evaporated into a gas cylinder and diluted with $N_2$ gas (Scott-Marrin). This $CH_3I$ mixture was further diluted with $N_2$ to a 5 ppm $CH_3I$ mixing ratio which was found to reproduce the field sensitivities of HP, MHP, and $H_2^{18}O$ observed in DC3 (Treadaway, 2015). The final reagent gas blend of $CH_3I$, $CO_2$, $O_2$, and $N_2$ yielded responses for $I^-$, $O_2^-$, and $O_2^-(CO_2)$ cluster ions with organic acids, peroxides, hydroxyacetaldehyde, and water vapor.

**2.4 Calibration Configuration**

HFo and HAc standards (HCOOH, > 95% and $CH_3COOH$, 99.9%, respectively) were obtained from Sigma-Aldrich. The HP standard was obtained from Fisher-Scientific ($H_2O_2$, 30%) and the MHP standard was synthesized (Lee et al., 1995a). Dilutions of both were standardized by titration and or UV absorbance (Lee et al., 1995a). In-flight calibrations were performed by microfluidic injection. Two versions of the microfluidic system were used to inject the liquid standard into the PCIMS via a nitrogen gas line. For the first set-up, used during DC3, the standard was contained in a Hamilton glass syringe and injected using a single syringe pump (1 x $10^{-6}$ L/min aqueous flow rate, KD Scientific Inc., Holliston, MA). The liquid standard was vaporized in a heating block (328 K) into a gaseous $N_2$ stream (0.4 slpm). The disadvantage of this system is that it can only calibrate for peroxides or organic acids and was used exclusively for the peroxides, as they were the target analytes of interest. After DC3, a second microfluidic system was developed which allowed for calibration of peroxides and organic acids. Both peroxide and organic acid aqueous standards (in Hamilton glass syringes) were injected (5 x $10^{-7}$ L/min) and evaporated into a

$N_2$ gas stream (0.4 slpm) via mixing-Ts and a ballast PFA-Teflon® mixing vessel. Both microfluidic standard addition systems were contained within the PCIMS instrument rack. In-flight calibrations were done as part of the FRAPPÉ program in the summer of 2014 with the second microfluidic set-up. During FRAPPÉ the organic acid aqueous standards were verified by titration (Treadaway, 2015). The percent errors between the theoretical and titrated concentrations were 1.00% and 1.51% for HFo and HAc, respectively. The FRAPPÉ peroxide aqueous standards, which were also used in post-mission laboratory work, were standardized by titration and/or UV absorbance with an estimated accuracy of 5% and 10%, respectively.

Sensitivities were determined in-flight by standard addition. The ambient signal before and after the calibration gas addition was used to estimate the ambient signal at the time of calibration gas addition. The sensitivity was then determined by dividing the calibration gas mixing ratio in the reaction cell by the difference between the combined standard addition and ambient signal and the interpolated ambient signal. The sensitivity of each compound is reported as counts per second per ppb (cps/ppb). The average error in laboratory sensitivity for HFo and HAc was 26% and 31% respectively. This accounts for error in the PCIMS signal response and error in instrumental sources (e.g. mass flow controllers).

Henry's Law constants were determined for HFo and HAc using a gas-aqueous coil equilibrium apparatus. HFo (0.3 mM) and HAc (0.9 mM), were acidified (0.02 N $H_2SO_4$) to keep each acid in its protonated form and thereby ensure partitioning into the gas phase according to each acid's Henry's Law constant. Henry's Law constants from Johnson et al. (1996) were used. Zero air (0.2 or 0.4 slpm) was passed through an equilibration coil in a water bath kept at 288 or 298 K along with the organic acid standard. The resulting calibration gas was added to the sample air stream after humidification (Sect. 2.6). For the work at 298 K, the laboratory room temperature was increased to 303 K to prevent water vapor from condensing on the transfer tubing walls. This same set-up was used for the GA Henry's Law experiment and the alcohol interference work described below.

PCIMS response and sensitivity to GA at m/z 92 ($O_2^-(GA)$) and m/z 187 ($I^-(GA)$) was determined using two different methods to generate known amounts of GA based upon the literature: 1) the Henry's Law constants of Betterton and Hoffmann (1988) and 2) the GA vapor pressure determination over neat GA melt as a function of melt temperature by Petitjean et al. (2010) with a serial gas dilution system. GA dimer was used as purchased (Sigma-Aldrich, St Louis, MO).

For the Henry's Law experiment, $3.689 \times 10^{-4}$ kg of GA dimer was dissolved into $1.00 \times 10^{-4}$ $m^3$ of pure water (18 Mohm), yielding a 0.0614 M solution of GA monomer. The same gas-aqueous equilibration coil apparatus was used as described above for the organic acid Henry's Law work. From the data of Betterton and Hoffmann (1988), the GA Henry's Law constant was predicted to equal 70 M hPa$^{-1}$ at 288 K. The direct application of this value to our experiments was referred to as Case 1. Betterton and Hoffmann noted their Henry's Law constants for GA were significantly larger than expected. Implicit assumptions in their analysis were the GA solution was all monomer (GA and GA hydrate) and aqueous hydration/dehydration kinetics were "fast" compared to the gas-aqueous equilibration time scale of their experimental system. However, Kua et al. (2013) reported that a 1 M GA monomer equivalent aqueous solution is a mixture of monomers and several dimer and trimer compounds. GA monomers were found to comprise approximately 55% of their solution with the monomer making up 3% and the monomer hydrate 52%. Using the experimental equilibrium constants determined from Kua et al. (2013) and our "as monomer" aqueous concentration, our aqueous solution was expected to be 91% monomer hydrate, 6% monomer with the remaining 3% nearly all dimer. Kua et al. also indicated the kinetics of the trimer and dimer equilibration was "slow," up to a few hours. Using these distributions and an assumption of "fast" monomer kinetics but "slow" kinetic exchange of trimer and dimer to monomer, the gas phase mixing ratio would be 97% of the reported Betterton and Hoffmann expected gas-phase mixing ratio at our aqueous equilibration concentration, referred to here as Case 2. Further, if the monomer hydration/dehydration kinetics were also "slow" such that the monomer hydrate does not have sufficient time in the equilibrator to convert to monomer (e.g., dehydration rates of Sørenson, 1972, are on the order of 0.01 to 0.1 s$^{-1}$ depending upon solution pH), then we would observe as little as 6% of the GA

gas as expected from the Betterton and Hoffmann (1988) Henry's Law constant and this situation was referred to as Case 3. The conditions of Case 2 and Case 3 would falsify the equilibrium assumption and cause the Betterton and Hoffmann Henry's Law constant to be too large as they noted. Table S2 in the supplemental information lists the expected reaction cell GA mixing ratio for these three cases at the five equilibration air flow rates used in the Henry's Law experiments. The GA sensitivity was
determined at two reaction cell water vapor mixing ratios, 1700 and 7500 ppm.

       In the melt "vapor pressure" GA source experiments, $1 \times 10^{-4}$ kg of GA dimer was placed in a $1 \times 10^{-5}$ m$^3$ glass vessel and slowly heated in a stirred water bath until fully melted at 358 K. A $1 \times 10^{-3}$ slpm flow of 532 ppm $CO_2$ in pure air was passed through the 10 mL vessel holding the melted dimer and the outflow immediately mixed with an Aadco air stream flowing at 0.3 slpm to prevent deposition of the GA monomer gas onto the walls of the vessel and gas transfer lines. The residence time of air
in the vessel was 10 minutes and sufficiently long to allow mixing of the air over the melt and for the melt to be in equilibrium with gas phase GA. The melt remained limpid as the bath temperature decreased to room temperature, nominally 295 K, and as the water bath was heated the next day up to a temperature of 358 K. The glass vessel and gas mixing-Ts were submerged in the water bath and the temperature was increased from 298 to 358 K in 20 K increments. The water bath temperature was monitored and this temperature was used to evaluate the partial pressure of GA above the melt. Table S3 shows the expected GA reaction
cell mixing ratio at different melt temperatures using the data from Petitjean et al. (2010).

       The potential exists for ethanol and 1-propanol or 2-propanol to be isobaric interferences in the measurement of HFo and HAc or GA, respectively, at the PCIMS m/z resolution of 1.0. The PCIMS sensitivity to these compounds was determined using their respective Henry's Law constants (Sander, 2015) and the gas-aqueous equilibration calibration apparatus described above. The alcohols were used as purchased (Sigma-Aldrich, St Louis, MO) and diluted with pure water to final concentrations
of $1 \times 10^{-4}$ M for ethanol, 1-propanol, and 2-propanol.

**2.5 Blank Configuration**

       Carulite-200® (Carus Corporation, Peru, IL), a magnesium dioxide/copper oxide catalyst, is an effective ozone and peroxide destruction catalyst and was used during DC3 as an analytical blank substrate for the peroxides (O'Sullivan et al.,
2018). It further proved to be effective in removing but not destroying the organic acids as well. Unfortunately, at low organic acid concentrations, there can be a positive trap response due to outgassing from the Carulite-200®. Therefore, three different traps were tested as organic acid blank substrates: $Cu/NaHCO_3$, $Na_2CO_3$, and NaOH. It was determined that the NaOH (5%) trap was effective at removing organic acids but not peroxides. Running the air sample through the Carulite 200® and then the NaOH trap removed both peroxides and organic acids with minor outgassing.
In flight, blanks were performed periodically. Field detection limits were determined from signal variability (3 times the standard deviation) during the trap-on cycle. The in-flight detection limits were 16 ppt for HFo and 50 ppt for AAES. In laboratory work, detection limits were calculated as three times the standard deviation of the Aadco background and are reported in Table 2 as a function of inlet pressure.

       In FRAPPÉ, the calibration and blank cycles were both 720 s in duration. The calibration gas was on for 75 s and off
for 645 s. The calibration gas was turned on coincident with the blank traps being turned off. The 16 selected m/z signals were sampled in 3.5 s. The full-response rise time and fall time for calibration gases on and off were 11 and 7 s, respectively for peroxides at m/z 80 and 110 and organic acids at m/z 173 and 187. The full-response fall time and rise time for the traps on and off were 14 and 11 s, respectively.

**2.6 Laboratory Set-Up**

The laboratory set-up was described in detail in Treadaway (2015) and only briefly presented here. In the laboratory, different field conditions were simulated by varying the water vapor and/or the inlet pressure of the sample air stream as depicted in Fig. 1b. A zero-air generator (Aadco Instruments Inc., Cleves, OH) supplied the sample air stream to prevent the addition of organics and excess water into the system. This air stream was split between "dry" and humidified lines. The dry line came directly from the Aadco.  The water concentration in the humidified line was controlled with two gas washing bottles and a gas-

water equilibration coil immersed in a water bath kept at 288 K or 298 K. By changing the ratio of air flow through the dry and humidified lines, it was possible to alter the overall water vapor mixing ratio in the air stream entering the PCIMS.  The inlet pressure was manually controlled after humidification with a needle valve (V, Fig. 1b) and a pressure transducer.  The needle valve was able to approximate the atmospheric altitude/pressure conditions (sea level to 14 km, approximately 120 hPa) experienced in the field and inlet pressure change impacts on signal response or sensitivity were investigated (Treadaway, 2015).

The reaction cell water vapor range, reagent gas reaction cell mixing ratios, and sample pressures used in the laboratory are given in Table 3.

**3 Results**

    A laboratory calibration mass spectrum (Fig. 2) highlights the $O_2^-$, $O_2^-(CO_2)$, and $I^-$ cluster signal responses for HP, MHP, HFo, and HAc in the multi-reagent ion system. For this scan, the dwell time at each mass was 50 milliseconds and the

ambient pressure was 1013 hPa, and the reaction cell water vapor mixing ratio was 370 ppm. PCIMS signal responses for HP include m/z 66 ($O_2^-$( HP)), m/z 110 ($O_2^-(CO_2)$(HP)) and m/z 161 ($I^-$(HP)). MHP is measured at m/z 80 ($O_2^-$(MHP)) and m/z 175 ($I^-$(MHP)). See O'Sullivan et al. (2018) and Heikes et al. (2017) for a more complete discussion of the ion cluster chemistry of HP and MHP.  HFo responds at m/z 78 ($O_2^-$(HFo)) and at m/z 173 ($I^-$(HFo)). HAc responds at m/z 92 ($O_2^-$(HAc)) and m/z 187 ($I^-$(HAc)) as does GA. The $I^-$ concentration in the PCIMS is monitored with the $I^-(H_2^{18}O)$ cluster (m/z 147). The $I^-$ signal in the

PCIMS (m/z 127) is marked as well for reference and under the reagent conditions saturates the detector; similarly the signal at 145 for $I^-(H_2O)$ was typically saturated as well.

    This blended reagent ion system hinges on a balance between the iodide and oxygen chemistry. In general, as the proportion of $CH_3I$ increased the sensitivity of the $CO_2$ and $O_2$ clusters decreased with the impact on MHP being greater than that for HP. The PCIMS is not as sensitive for HAc as for HFo (Figs. S1, 3, 4) and a sufficient amount of $CH_3I$ is needed to promote

HAc clustering. Therefore, finding a balance between the two reagent gases ultimately depends on a prioritization between MHP and HAc. For this reason, five $CH_3I$ flow rates (0.0005, 0.001, 0.0015, 0.002, and 0.0025 slpm) were evaluated. Figure S1 shows $I^-$ cluster laboratory sensitivities for $I^-$(HFo), $I^-$(HAc), $I^-$(HP), and $I^-$(MHP) as a function of $CH_3I$ flowrate. Figure S2 shows the laboratory MHP sensitivity at m/z 80 ($O_2^-$(MHP)) as a function of $CH_3I$ flowrate. All of the pressure and water work is combined together which accounts for the large variance shown (1 standard deviation). The ion clusters' water dependencies are discussed

below. As the $CH_3I$ flowrate increased, the $O_2^-$(MHP) sensitivity decreased. As expected, the sensitivities of the $I^-$(HFo), $I^-$(HAc), $I^-$(HP), and $I^-$(MHP) clusters increased as the $CH_3I$ flowrate increased with an approximate doubling in sensitivity for HFo and HP corresponding with a doubling in $CH_3I$ flowrate. Overall an increase in $CH_3I$, and consequently $I^-$, resulted in an increase in $I^-$(HAc) sensitivity but at the cost of decreasing the $O_2^-$(MHP) sensitivity. It was fortuitous that there was enough $CH_3I$ present during DC3 to promote organic acid clustering without impairing the $O_2^-$(MHP) sensitivity. The data of Fig. S3, $I^-$(HP) in Fig. 3,

and those for $O_2^-$(HP), $O_2^-(CO_2)$(HP) (not shown) were used to identify the $CH_3I$ flow rate of 0.0005 slpm as providing the best sensitivity matches to the DC3 calibration data for HP and MHP.

Figure S3 shows the MHP calibrations at m/z 80 ($O_2^-$(MHP)) from DC3 as a function of reaction cell water vapor mixing ratio. Laboratory derived MHP sensitivity at m/z 80 is also shown as a function of reaction cell water vapor mixing ratio for 5 different $CH_3I$ flow rates. The data are binned by the reaction cell water vapor mixing ratio. The mean sensitivity for that bin is plotted and the horizontal bar represents the limits of the reaction cell water vapor mixing ratio. The length of the vertical bar from the mean represents one standard deviation and includes random errors associated with variations in pressure, ambient concentrations during the standard addition, and systematic variations due to water vapor in a bin, calibration gas precision, and instrumental precision. Figure 3 shows $I^-$ cluster sensitivities for $I^-$(HP), $I^-$(MHP), $I^-$(HFo), and $I^-$(HAc) for the FRAPPÉ experiment and from the same $CH_3I$ laboratory work as in Fig. S3. The horizontal and vertical error bars represent the same information as in Fig. S3.

The laboratory calibration technique was verified by comparison to in-flight calibrations from FRAPPÉ. The in-flight FRAPPÉ calibrations are included in Fig. 3. The first two FRAPPÉ flights are omitted due to in-flight vibrations (Heikes et al., 2017). HAc calibrations were not available for all flights due to contamination issues in the hanger (Heikes et al., 2017) and vibration. The vibration problem led to "chatter" in the mass flow controllers and their orientation and location within the instrument rack was modified between flights several times. The HFo and HAc laboratory sensitivities were similar to the FRAPPÉ in-flight calibrations. HAc sensitivity decreased with water above 1000 ppm. The HP and MHP FRAPPÉ sensitivity averages were higher than the 0.0005 slpm laboratory work but within the error. $I^-$(MHP) was independent of water but there appeared to be a water sensitivity maximum for $I^-$(HP) at about 1000 ppm reaction cell water vapor. There was a pressure dependency in the sensitivity of $I^-$(HFo) and $I^-$(HAc); however, it was found insignificant compared to the dependence with water vapor and is not discussed further. Treadaway (2015) contains a complete analysis of the pressure dependency investigation.

FRAPPÉ in-flight sensitivities as a function of reaction cell water vapor for PCIMS analyte clusters are shown in Fig. 4. The horizontal and vertical error bars represent the same information as in Figs. S3 and 3. Figure 4a contains the $O_2^-$ cluster calibration data for HP, MHP, HFo, and HAc. The $O_2^-(CO_2)$(HP) cluster is also included on Fig. 4a. $O_2^-$(HAc) sensitivity was independent of water vapor but the other four compound sensitivities decreased with increasing water vapor over the range of reaction cell water vapor mixing ratios observed in FRAPPÉ. Figure 4b shows the $I^-$ cluster sensitivities for HP, MHP, HFo, and HAc. As described above, the $I^-$(HP) and $I^-$(HAc) sensitivities decreased with water vapor mixing ratio whereas $I^-$(HFo) and $I^-$(MHP) increased with reaction cell water vapor mixing ratio.

Henry's Law constants were determined for HFo and HAc at 288 and 298 K and are presented in Table 4 along with the reaction enthalpies. A wide range of Henry's Law constants from 5.4 to 13 M/hPa and 5.4 to 9.2 M/hPa have been reported for HFo and HAc at 298 K, respectively (Sander, 2015). Of the measured values reported in Sander (2015), only Johnson et al. (1996) experimentally determined the Henry's Law constants at multiple temperatures. Our Henry's Law constants compared best to those given by Johnson et al. (1996), especially for HAc. The Henry's Law constants for HFo were lower than the Johnson et al. (1996) values. The difference in Henry's Law constants could be due to a higher gas-phase partitioning through the coil system than measured by Johnson et al. (1996). Our reaction enthalpies for HFo were higher than the Johnson values which also could be due to a higher gas-phase partitioning in our system. The HAc smaller reaction enthalpy, relative to Johnson's value, was likely due to the higher Henry's Law constant for HAc at 298 K. It is the only value in our work that is higher than Johnson. It is possible that at the higher temperature, and therefore higher water vapor mixing ratio in the reaction cell (Treadaway, 2015), we were actually seeing a decrease in HAc sensitivity not captured in the laboratory syringe calibrations that occurred at lower water vapor mixing ratios. This would have caused us to overestimate our Henry's Law constant.

Ethanol (hereafter referred to as EtOH), 1- and 2-propanol (hereafter referred to as 1- and 2-PrOH), and glycolaldehyde (GA) are potential isobaric interferences for $I^-$(HFo) and $I^-$(HAc). The PCIMS sensitivity to $I^-$(EtOH), $I^-$(1-PrOH), and $I^-$(2-PrOH)

was quantified using the Henry's Law equilibration system. The PCIMS was substantially more sensitive to HFo and HAc compared to these alcohols. At the lowest tested reaction cell water vapor mixing ratio (~30 ppm), the PCIMS was 140 times more sensitive to HFo compared to EtOH and the ratio increased with increased water vapor mixing ratio. At the lowest reaction

cell water vapor mixing ratio, the PCIMS HAc sensitivity was 140 and 90 times those for 1- and 2-PrOH, respectively. As with the EtOH measurements, the sensitivity to HAc relative to 1- and 2-PrOH increased with increasing reaction cell water vapor mixing. Baasandorj et al. (2015) performed a similar study for EtOH and 2-PrOH using a PTR-MS instrument and reaction cell water vapor range equivalent to 2500 – 15000 ppm. They found the HFo sensitivity to be 6 to 15 times higher than that for EtOH and their HAc sensitivity was 200 – 300 times higher than that for 2-PrOH over their experimental humidity range. It should be

acknowledged that these two techniques are different and some of the masses detected by the PTR-MS were fragments of the alcohols. While a time-of-flight CIMS can distinguish the alcohols from the organic acids (Yuan et al., 2016), there is a paucity of quadruple I⁻ CIMS data available with which to compare our I⁻ CIMS alcohol interference work.

The PCIMS sensitivity to GA was evaluated using a Henry's Law equilibration system and a vapor pressure melt system to generate gaseous GA and the results are presented in Tables 5 and 6, respectively. The sensitivities for the two GA generation

systems were further compared to the HAc sensitivity (Table 5; comparison sensitivity was developed from Fig. 3 and Fig. 4). Case 1 and Case 2 are reported together because the sensitivities were indistinguishable for reportable significant digits; therefore, comparison to the melt method and HAc only considered Case 1 or 3. The GA sensitivities at m/z 92 ($O_2^-$(GA)) and m/z 187 (I⁻(GA)) for the melt vapor pressure source of GA were between those from the Case 1 and Case 3 assumption sets for the Henry's Law generated GA sensitivities. The GA sensitivities using the Case 1 assumptions were comparable to the HAc

sensitivity at m/z 92 and m/z 187. The GA sensitivities determined using the melt vapor pressure source were a factor of 4 and a factor of 10 greater than the sensitivity of HAc at m/z 92 and m/z 187, respectively. Unlike Petitjean et al. (2010), we did not purify the GA dimer using a freeze-pump-thaw cycle. This could have led to potential impurities in the solid, one of which could be HAc, and possibly an overestimation of the vapor pressure. Magneron et al. (2005) also reported partial pressure ranges for GA at 298 and 333 K and the value at 298 K was 20 times higher than Petitjean et al. Petitjean et al. (2010) suggested that this

difference could be from volatile impurities. If we use the Magneron et al. vapor pressures instead of Petitjean et al. our sensitivities at 298 K were $1 \times 10^4$ and $1 \times 10^3$ cps/ppb for m/z 92 and m/z 187, respectively. These sensitivities are substantially closer to the gas-aqueous work from Case 1. The GA reaction cell mixing ratio of GA using Magneron's vapor pressure values were 22 ppb at 298 K and 64 ppb at 333 K (we measured at 338 K). In comparison, using Petitjean's vapor pressures the GA reaction cell mixing ratios were 2 ppb and 39 ppb at 298 K and 338 K, respectively. Our high sensitivities determined with the

Petitjean et al. vapor pressures could be due to impurities in the sample. Regardless, these results imply GA or HAc were a significant interference in the measurement of the other using both $O_2^-$ and I⁻ cluster formation. As GA atmospheric mixing ratios are non-negligible (Table S1), PCIMS data collected at m/z 187 are reported as the "acetic acid equivalent sum," or AAES, of HAc plus GA.

## 4 Discussion

### 4.1 Ion Chemistry and Water Sensitivity Dependence

Jones et al. (2014), Le Breton et al. (2012), and Lee et al. (2014) observed an I⁻(HFo) sensitivity dependence on water vapor. Lee et al. (2014) has shown I⁻(HAc) sensitivity to vary with water vapor. O'Sullivan et al. (2018) and Heikes et al. (2017) discussed the water sensitivity of $O_2^-$($CO_2$)(HP) and $O_2^-$(MHP) clusters. HFo and HAc sensitivities were the primary focus of this work and were examined over a range of water vapor mixing ratios from ~30 ppm to 20,000 ppm with a combination of

laboratory and field measurements. I⁻(HP) sensitivity was also examined as it was used together with I⁻(H₂O), O₂⁻(CO₂)(HP), and O₂⁻(MHP) sensitivities to diagnose the PCIMS residual CH₃I mixing ratio present in DC3. In addition, a weak MHP calibration signal at m/z 175 was observed in FRAPPÉ. Heikes et al. (2017) used these data and developed a more detailed analysis of the I⁻ chemistry of HFo, HAc, HP, and MHP, which is briefly presented below.

The following ion chemistry was invoked to account for an iodide cluster's observed sensitivity dependence on water vapor (Lee et al., 2014; Heikes et al., 2017)

$$(1) \qquad\qquad I^- + H_2O + M \rightarrow I^-(H_2O) + M$$

$$(2) \qquad\qquad I^- + X + M \rightarrow I^-(X) + M$$

$$(3) \qquad\qquad I^-(H_2O)_n + X \leftrightarrow I^-(X)(H_2O)_{n-1} + H_2O$$

$$(4) \qquad\qquad I^-(X) + H_2O \leftrightarrow I^-(H_2O) + X$$

where X represents HFo, HAc, HP, and MHP and M represents a third-body reactant (typically N₂, O₂, H₂O, and CO₂). Heikes et al. found that the pressure and humidity trends seen in our PCIMS laboratory and field work for HP, HFo, and HAc could not be replicated without the addition of I⁻(H₂O)₂ (3), especially at the higher humidity values. However, I⁻(H₂O)₂ was not present in mass scans in FRAPPÉ or the laboratory and we inferred I⁻(H₂O)₂ binding was not strong enough to survive declustering in the collision dissociation chamber.

Lee et al (2014) found the I⁻(HFo) sensitivity plateaus and declines when the reaction cell water was above 2200 ppm. The occurrence of a maximum sensitivity as a function of water vapor is two-fold. First, Iyer et al. (2016) and Heikes et al. (2017) have pointed out the rates of cluster forming reactions (2) are promoted by a third-body reactant which acts as an energy carrier and stabilizes the cluster. H₂O is expected to be more efficient in this regard than the other molecules listed above. Second, H₂O competes with X for I⁻ (1) and can shift the switching reaction (4) equilibrium in favor of I⁻(H₂O) thereby decreasing the yield of I⁻(X) when H₂O is large. Unlike Lee et al. (2014), our HFo sensitivity did not decrease at the highest water mixing ratios tested, though it appeared to plateau - most notably in the ambient pressure (1013 hPa) laboratory work (Fig. 3). Possibly, our highest reaction cell water mixing ratios were insufficient to achieve a decline in sensitivity as observed by Lee et al. The maximum water mixing ratio in the reaction cell during laboratory experiments was 7800 ppm (Treadaway, 2015). However, the FRAPPÉ in-flight calibrations covered a larger water mixing ratio yet there was still no decline in sensitivity (Fig. 4). It is likely that instrumental differences between the two CIMS configurations led to a shift in the location of the water response peak in sensitivity. Lee et al. (2014) used a much higher CH₃I reagent gas mixing ratio and reaction cell pressure (90 hPa) or [M] which, as mentioned above, can impact the reaction velocity (1, 2). Jones et al. (2014) and Le Breton et al. (2012) intentionally added water to promote clustering. Jones et al. (2014) found a decrease in sensitivity at their lowest water mixing ratios as a result of an insufficient water source to promote clustering under the dry sampling conditions of the Arctic and upper troposphere. Under the Le Breton et al. sampling conditions near the surface they operated in a water vapor independent regime. Our in-flight observations and unpublished Heikes et al. (2017) model results with Le Breton's CH₃I mixing ratio suggests that there is a water dependent regime between the altitudes sampled by Jones et al. and Le Breton et al.

Figures 3 and 4 show I⁻(HAc) sensitivity was constant up to approximately 1000 ppm reaction cell water vapor mixing ratio, above which the sensitivity decreased. This suggested reaction (2) for HAc was likely able to dissipate the excess energy of reaction into the cluster ion without requiring an explicit third body molecule. Above 1000 ppm, I⁻(HAc) sensitivity decreased with increasing reaction cell water vapor mixing ratio, indicating the switching reaction equilibrium for HAc (4) behaved like

what was expected for HFo, but not observed, and was shifting towards $I^-(H_2O)$. By comparison, Lee et al. (2014) found a decrease in $I^-(HAc)$ sensitivity with the addition of any water to their system.

Iyer et al. (2016) reported a binding enthalpy of -70.5 kJ/mol for $I^-(HAc)$ and -106.8 kJ/mol for $I^-(HFo)$. The binding enthalpies are reported here as negative values, indicating an exothermic process and opposite to the NIST nomenclature for ion-molecule reactions (Bartmess, 2017). They correlated the sensitivities of Lee et al. to binding energy and theorized the binding enthalpy for an analyte in an $I^-$ cluster could be used to predict its sensitivity. Figures 3 and 4 suggested ambient water vapor also had a significant role to play in determining an analyte's sensitivity with our $I^-$ CIMS configuration.

**4.3 Interferences**

HFo, HAc, and GA were found to form cluster ions with both $O_2^-$ and $I^-$ ions. Figure 4, developed from FRAPPÉ data, demonstrated the $O_2^-$ cluster sensitivity for each of the analytes was greater than its $I^-$ counterpart. By itself this argued for the use of $O_2^-$ over $I^-$. However, m/z 78 ($O_2^-(HFo)$) in our system may experience interference from cluster ions such as $CO_3^-(H_2O)$ and $^{18}O$ of $O_2^-(CO_2)$ also at m/z 78 (Heikes et al., 2017; O'Sullivan et al., 2018). Interference at m/z 92 ($O_2^-(HAc)$) included HAc interference by GA and vice versa and speculative cluster ions like $CO_3^-(O_2)$ or $NO_2^-(HFo)$. A second drawback to the use of $O_2^-$ as a cluster ion stems not from potential interferences but from the complex interplay between $O_2^-$, $CO_2$, and $H_2O$ and the analytes HP, MHP, HFo, HAc and GA (Heikes et al., 2017). Calibration under variable water vapor conditions and variable trace species such as ozone or nitrogen oxides was challenging.

From the results, it was clear HAc and GA provided comparable response as $O_2^-$ clusters or $I^-$ clusters, even though the GA gas phase Henry's Law and melt vapor pressure systems used here were not ideal as outlined above. The HAc:GA relative sensitivity was between 1:1 to 1:10. We are most confident in our Case 1 and Case 2 Henry's Law work which presumed "fast" monomer hydration/dehydration (both Case 1 and 2) and "fast" monomer, dimer, and trimer equilibrations (Case 1). To rule out "slow" dehydration/hydration equilibration kinetics (Case 3) in the GA aqueous solution, multiple gas flow rates through the coil were used. A "slow" dehydration of monomer was expected to result in a reduction in sensitivity as the flow rate was increased and monomer was depleted before replacement could occur from the monomer-hydrate pool. This was not observed and the hydration/dehydration kinetics were taken to be "fast". A Case 1 (or Case 2) result interpretation yielded a 1:1 sensitivity ratio and implies reported AAES mixing ratios were close to the true sum of HAc and GA. If the melt vapor pressure source sensitivity was correct, then we observed approximately a factor of 10 higher sensitivity for GA than for HAc. This implies reported AAES mixing ratios represent an upper limit to the sum of HAc and GA, and if in fact the AAES included only GA, the AAES indicates 10 times the amount of GA than actually present. Baasandorj et al. (2015) also tested GA interference in their PTR-MS HAc measurements. They found a HAc:GA sensitivity ratio of 0.65 – 1.4 over their experimental humidity range. Our Case 1 Henry's Law results, using drastically different ion chemistry, are consistent with their work. St. Clair et al. (2014) measured HAc and GA with both a single quadrupole and tandem CIMS with a $CF_3O^-$ reagent ion. Their single quadrupole HAc:GA ratio was 2:3 to 3:2 for four flights during the California portion of the Arctic Research of the Composition of the Troposphere from Aircraft and Satellites (CARB-ARCTAS). These flights sampled biomass burning and high biogenic emissions with urban influence from Sacramento (St. Clair et al., 2014). St. Clair's single quadrupole CIMS is similar to ours, though with a different reagent ion, and they also found a HAc:GA ratio consistent with our Case 1 Henry's Law results. As a caveat, the Petitjean et al. (2010) critique of prior work regarding GA absolute vapor pressure could apply to Baasandorj et al. (2015), St. Clair et al. (2014), as well as, our work and GA gas calibration is an unresolved issue.

**4.4 FRAPPÉ Example flight**

Figure 5 shows PCIMS HFo and AAES data from FRAPPÉ Research Flight 12 (RF 12) on August 12, 2014. The C130 flew a mountain-valley flight pattern to sample "upslope" flow over the Rocky Mountains. Part 1 of the flight was flown between Boulder and Greeley in a series of stacked legs. Part 2 (after refueling at 16:00 MDT) flew over Denver and then two legs over the Continental Divide with a low altitude "missed approach" at Granby airport on the western side of the divide. Both HFo and AAES mixing ratios were at least 1 ppb for the majority of the flight. The highest HFo was found west of Fort Collins near biogenic sources characterized by isoprene greater than 75 ppt, methyl vinyl ketone (MVK) greater than 100 ppt, and methacrolein (MACR) greater than 70 ppt (NCAR Trace Organic Gas Analyzer, Apel et al., 2015). Elevated HFo (>1.5 ppb) in Granby corresponded with elevated $O_3$ (~80 ppb, NCAR 1-channel chemiluminescence, Ridley et al., 1992) and a biogenic signature (~100 ppt MVK and ~80 ppt isoprene). This could be secondary production from an upslope flow event, and subsequent spill over event (Pfister et al., 2017). There was high AAES (up to 14 ppb) below 0.5 km (AGL, above ground level) corresponding with high $NH_3$ (Aerodyne Research, Inc., Herndon et al., 2005) with a maximum mixing ratio of 180 ppb near Greeley which is an area associated with a concentration of confined animal feedlot operations (Eilerman et al., 2016; Yuan et al., 2017). If the signal at m/z 187 were primarily HAc, the $HAc:NH_3$ ratio was 0.078 ppb/ppb which is within the range reported by Paulot et al. (2011) though larger than the enhancement ratio range of 0.02-0.04 ppb/ppb reported by Yuan et al. (2017). A maximum AAES of ~10 ppb was measured over the Denver Metropolitan area, when HFo was approximately 1 ppb.

**4.5 DC3 Vertical Profiles and Test Case**

The DC3 observations were divided into three study regions as indicated by the colored boxes in Fig. 6a and labeled Colorado-Nebraska, Oklahoma-Texas, and Eastern region (states from Arkansas to the Carolinas). HFo and AAES data for the three sub-domains were composited as a function of altitude and the composite profiles are shown in Fig. 6b-6d. The measurements are binned in 1 km intervals, where the symbols denote the bin median value, the thicker lines indicate the bin inner-quartile range, and the thin lines show the $10^{th}$ to $90^{th}$ percentile range. Stratospherically influenced air was removed before bin statistics were computed by eliminating air samples with high ozone (> 150 ppb) and low carbon monoxide (< 70 ppb).

Each study region had lower HFo mixing ratios compared to AAES. Previous field measurements reported varied results about the proportion of HFo to HAc. Reiner et al. (1999) and Talbot et al. (1996) reported less HFo relative to HAc (by as much as a factor of 2 from 7-12 km). Millet et al. (2015) sampled HFo and HAc during the summer over the US Southeast and found the mean HFo to HAc ratio to be 1:1 at their maximum reported altitude (approximately 5 km) and 1.0:1.4 at the lowest near surface altitudes. Millet et al.'s HFo mixing ratios were an order of magnitude higher than reported here though our AAES mixing ratios were within Millet et al.'s reported HAc mean plus/minus standard deviation range. The high solubility of HFo and the large extent of vertical mixing characteristic of the stormy conditions sampled during DC3 likely led to a preferential sampling of conditions that diluted, and possibly wet-deposited, HFo. These same conditions would also lead to diluted and scavenged AAES measurements if AAES was mostly composed of GA.

In general, all three profiles had a decrease in HFo up to 6 km followed by an increase back to boundary layer mixing ratio values or higher. This profile was most pronounced in the Eastern DC3 region. The Eastern region also had the highest-altitude measurements and the HFo sensitivity started to decrease again above 12 km. The highest mixing ratios of both HFo and AAES in the Oklahoma-Texas region were measured at 2 km. The Colorado HFo profile has more HFo at the top of the profile than in the boundary layer. The AAES altitude trend was not as strong in any of the study regions though the mixing ratio decreased up to 6 km. The Eastern region had the biggest difference between both HFo and AAES at high altitude. The largest

range of mixing ratios (represented by the 10th-90th percentile) was in the Oklahoma-Texas region and was reflected in both the peroxide (not shown) and HFo/AAES profiles.

Figure 7 shows HP, MHP, HFo and AAES mixing ratios during DC3 Research Flight 5. The HIAPER altitude is plotted as well for reference. The mission was to sample convective outflow from a Texas/Oklahoma storm the night before. During a low altitude leg, HFo was approximately 400 ppt and AAES was ~1400 ppt in a biogenically active area rich in isoprene, ~6 ppb (NCAR Trace Organic Gas Analyzer, Apel et al., 2015). AAES was greater than HFo during most of the flight. The HIAPER also sampled biomass burning during this flight (indicated on Fig. 7). AAES was >1 ppb during biomass burning sampling. Biomass burning was identified by a CO enhancement of 80 ppb and HCN enhancement of >200 ppt above background. There is no MHP reported during this period due to potential interferences at mass 80 from $CO_3^-(H_2O)$ with an $^{18}O$ and/or $NO_3^-(H_2O)$ (Heikes et al., 2017). The storm outflow portion (identified by MHP>HP) had periods of elevated HFo (~400 ppt) similar to the low altitude measurements earlier in the flight. A comparable increase back to lower altitude mixing ratios was not seen in AAES. Based on effective Henry's Law constants and retention factors (e.g. Barth et al., 2007), HAc is expected to be more efficiently transported through such storms relative to HFo and therefore expected to have a greater mixing ratio in the storm outflow. If AAES was dominated by GA, the expected outflow AAES would be lower than HFo given the higher Henry's Law constant of GA. The AAES mixing ratio in the storm outflow was about 2-3 times lower than in the biomass burning plume; however, it was greater than the HFo which suggested AAES was likely a more balanced sum of HAc and GA and not dominated by GA.

We have attempted to examine our AAES data in light of prior measurements of GA and HAc in biogenic or isoprene rich air masses, biomass burning plumes, and urban areas. Lee et al. (1995b, 1998) reported GA surface and aircraft measurements from the Southern Oxidation Study at a rural Georgia surface site in July and August 1991 and in June 1992 and from aircraft measurements from the Nashville/Middle Tennessee Ozone Study conducted in June and July 1995. They did not measure or report HAc. HAc aircraft data were compiled by Khare et al. (1999) and tower observations made by Talbot et al. (1995) (Shenandoah National Park, September 1990). Combining these datasets, a surface HAc:GA ratio ranged from 0.9 to 10 and the aircraft ratio, using HAc from remote regions, was from 1 to 14. Convolving our Case 1 HAc and GA relative sensitivities (1:1) and the synthetic ratios from these four data sources, an AAES value of 2 ppb would represent anywhere from 1 ppb of both HAc and GA to 1.9 ppb HAc and 0.13 ppb GA. Doing the same with our vapor pressure determined response ratio of 1:10, then the same AAES value of 2 ppb would represent HAc and GA mixing ratios from 0.17 and 0.18 ppb to 1.2 and 0.083 ppb, respectively. As seen above, in biogenically dominated areas it is possible to have 1:1 proportions of HAc to GA in the AAES measurements but HAc would dominate at the higher reported HAc mixing ratios.

GA, HAc, and HFo should be co-emitted in fires. Biomass burning is a primary emitter for GA and HAc and secondary for HFo (Khare et al., 1999; Yokelson et al., 1997, 2009). Using summary data from Akagi et al. (2011) and Stockwell et al. (2015) on emission ratios and emission factors, it is reasonable to expect enhancements of 20-30 ppt in HFo, 170-180 ppt in HAc, and potentially 30-40 ppt GA, for every 10 ppb enhancement in CO near the source for a North American biomass burning plume. St. Clair et al. (2014) found a higher average GA enhancement of 57 ppt for every 10 ppb enhancement in CO for both fresh and aged plumes. Performing the same analysis as above, we can estimate the proportion of HAc and GA from an AAES value of 2 ppb and a 10 ppb enhancement in CO. Based on the Case 1 Henry's Law HAc to GA relative sensitivities (1:1) and the enhancements reported above, there would be 1.67 ppb HAc and 0.33 ppb GA or, for the work of St. Clair et al., 1.5 ppb HAc and 0.5 ppb GA. Using the vapor pressure response ratio of 1:10, the same AAES value of 2 ppb per 10 ppb of CO would result in HAc and GA mixing ratios of 0.67 and 0.133 ppb, or 0.46 and 0.15 ppb for GA enhancement found by St. Clair et al,

respectively. We would expect that most of the AAES emitted from biomass burning would be HAc even at the 1:10 response rate because 3-5 times more HAc relative to GA is released.

There are limited measurements for GA in urban environments. Spaulding et al. (2003) and St. Clair et al. (2014) measured GA at a tower near the Blodgett Research Station on the western slope of the Sierra Nevada mountains. Spaulding et al. measurements were made in August and September 2000 and GA ranged from 0.092 – 1.7 ppb. St. Clair et al. measurements were made in June and July 2009 and they observed an average of 0.986 ppb and a maximum of about 4 ppb. This site is influenced by urban emissions from Sacramento and Spaulding et al. estimated 40% of the GA was attributable to anthropogenic origins. Therefore, we used 40% of the average GA reported by Spaulding et al. and St. Clair et al. for an urban estimate. Okuzawa et al. (2007) observed a maximum GA mixing ratio of 1.77 ppb in Tokyo. This is compared to our urban estimate from the Blodgett Research Station. Grosjean (1990) measured HAc in Southern California where it ranged from 0.9-13.4 ppb. From these studies we inferred an urban HAc to GA ratio between 3:2 and 49:1. Again taking a representative AAES value of 2 ppb, for the Case 1 scenario (1:1) there could be HAc and GA values anywhere from 1.39 ppb HAc and 0.61 ppb GA to 1.96 ppb HAc and 0.04 ppb GA for the minimum and maximum reported HAc values, respectively. Using the maximum HAc and GA reported mixing ratios to determine their ratio, there would be 1.77 ppb HAc and 0.23 ppb GA for an AAES value of 2 ppb. However if we use our vapor pressure determined HAc to GA response ratio of 1:10 and an AAES signal of 2 ppb, the HAc and GA ranged from 1.66 and 0.034 ppb to 0.374 and 0.16 ppb, respectively, for the Sacramento conditions. For the urban maxima, HAc and GA would be 0.9 and 0.11 ppb, respectively. Based on this analysis there would be at least twice as much HAc as GA measured as AAES in an urban air mass.

There is a continued need for simultaneous measurements of HAc and GA in urban to biomass burning to rural environments from the surface to upper troposphere. Baasandorj et al. (2015) developed a trap that removed HAc allowing GA to be measured by PTR-MS. We have not yet tested how effectively our current trap system removes GA and this also will need to be considered when reporting AAES results. We plan to develop a trap that will remove GA but leave HAc. With a dual trap system, it is conceivable HAc and GA can be determined sequentially and independently of each other using I⁻ CIMS or PTR-MS.

**5 Conclusions**

This study outlines the development of an airborne mixed reagent system to measure HP, MHP, HFo, and the acetic acid equivalent sum of HAc and GA. This is the first CIMS system to utilize simultaneous $O_2^-$, $O_2^-(CO_2)$, and I⁻ ion chemistry and was initially deployed in the field during FRAPPÉ and unintentionally deployed in DC3 when the focus was on HP and MHP alone. Ethanol, propanol, and glycolaldehyde, three isobaric interferences, were evaluated. Ethanol and 1- and 2-propanol were found be insignificant in the measurement of HFo at m/z 173 and in the measurement of HAc at m/z 187, respectively, unless the alcohol mixing ratio greatly exceeds the acid mixing ratio by a factor of ~20 or more. On the other hand, we found the PCIMS response to GA to be comparable to or greater than the instrument response to HAc. Consequently, HAc and GA have the potential to significantly interfere in the measurement of one another at both m/z 92 and 187. Given this result, our work with the PCIMS must report data collected at m/z 92 or m/z 187 as the "acetic acid equivalent sum" of HAc and GA, which is referred to as AAES. The post DC3 laboratory calibrations and deployment during FRAPPÉ permitted the quantification of HFo and AAES measured during DC3. All three DC3 study regions were characterized by greater AAES relative to HFo throughout the altitude profile and both organic acids had a "C" shaped altitude profile for the majority of the flights consistent with the deep convective transport of these species or their precursors. Future work will develop a new acid trap based on Baasandorj et al.

(2015) that removes HAc while leaving GA and conversely developing a trap which removes GA while leaving HAc. This will make it possible to measure each independently of the other.

**Acknowledgements:** This research was supported through grants from the United States National Science Foundation: ATM-09222886 and ATM-1063467 (DWO) and ATM-1063463 (BGH) and a contract from the Colorado Department of Public Health and the Environment (BGH).  The authors thank D. Tanner, G. Huey, and R. Stickle (THS Instruments, LLC) for advice and support developing our CIMS techniques. We thank the members of the DC3/SEAC4RS and FRAPPÉ/DISCOVER-AQ science teams and thank the NCAR-EOL Research Aviation Facility flight crew and staff for the quality of their data and for making the flight programs successful. Disclaimer: The manuscript has not been reviewed by the Colorado Department of Public Health and Environment. The scientific results and conclusions, as well as any views or opinions expressed herein, are those of the author(s) and do not necessarily reflect the views of the Department or the State of Colorado.

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

**Table 1. PCIMS instrument settings: voltages, pressures, temperatures, and MFCs set points**

| Description | Set Point / Nominal Value | Range |
|---|---|---|
| *Mass Flow Controller (MFC)* | | |
| $N_2$[1] Reagent for P control (MFC 3) | variable | ~2 to 4.6 slpm[2] |
| $CO_2$ in Air[3] Reagent (MFC 2) | 0.08 slpm[3] | |
| $CH_3I$ in $N_2$[4] Reagent (MFC 1) | 0.0005 slpm | ~0 to 0.01 slpm |
| $N_2$[1] Calibration Gas Carrier (MFC 4) | 0.4 slpm | |
| Inlet Excess Sample Flow (MFC 6) | 4.8 slpm | 3.6 to 5 slpm |
| Drawback Flow Calibration Gas (MFC 5) | 1.2 slpm | |
| *Pressure* | | |
| RXN Cell | 22 hPa | |
| CDC Chamber | 0.61 hPa | |
| Octopole Chamber | 0.0065 hPa | |
| QMS Chamber | 0.00011 hPa | |
| *Temperature* | | |
| HIML Inlet (FRAPPÉ / DC3) | 35 °C / 70 °C | |
| Inlet Transfer Line (FRAPPÉ / DC3) | 35 °C / 70 °C | |
| Liquid-to-Gas Tee (FRAPPÉ / DC3) | 45 °C / 55 °C | |
| *CIMS Instrument Voltages* | | |
| CDC Plate | 7 V | |
| CDC DC Bias | 20 V | |
| CDC RF | 2.0 V | |
| Octopole DC Bias | -0.04 | |
| Octopole RF | 2.49 | |
| Rear Ion Detector HV1 | 3.43 kV | |
| Front Ion Detector HV2 | 1.51 kV | |

[1]$N_2$ for RXN pressure control and calibration carrier gas was ultra-high purity nitrogen (Scott-Marrin) in FRAPPÉ and DC3 and liquid nitrogen boil off gas in the laboratory (Air Gas).

[2] slpm, standard liters per minute ($T_{ref}$ = 273.15 K; $P_{ref}$ = 1013.25 hPa).

[3]$CO_2$ (400 ppm) in ultrapure air (Scott-Marrin).

[4]$CH_3I$ (5 ppm) in ultrahigh purity $N_2$ (Scott-Marrin)



**Table 2: Laboratory detection limits (ppt) determined as three times the standard deviation of the blank using a pure air system as a function of sample inlet pressure (hPa)**

| Pressure, hPa | HFo, ppt | HAc, ppt |
|---|---|---|
| 120 | 46 | 86 |
| 180 | 23 | 46 |
| 306 | 13 | 37 |
| 600 | 18 | 59 |
| 1013 | 59 | 120 |







**Table 3: Laboratory instrument calibration conditions: sample inlet pressure, reaction cell water vapor mixing ratios, and reagent gas reaction cell mixing ratios**

| Sample Pressure, hPa | Reaction Cell Water Vapor Mixing Ratio[1], ppm | | Reaction Cell Reagent Gas Mixing Ratio | | | |
|---|---|---|---|---|---|---|
| | Low | High | $CH_3I$, ppb | $CO_2$, ppm | $O_2$, ppm | $N_2$, ppm |
| 120 | 40 | 540 | 0.575 | 7.36 | 3678 | 996322 |
| 180 | 50 | 610 | 0.580 | 7.42 | 3712 | 996288 |
| 306 | 90 | 1100 | 0.616 | 7.88 | 3941 | 996059 |
| 600 | 230 | 4400[2] | 0.814 | 10.42 | 5212 | 994788 |
| 1013 | 370 | 7700[2] | 1.174 | 15.02 | 7512 | 992488 |

[1]This work was performed with a water bath at 288 K

[2]This includes work in a water bath at 298 K

**Table 4: Henry Law constants and enthalpies for formic and acetic acid**

| Species | Temperature, K | $K_H$ This Work, M/hPa | $K_H$ Johnson *et al.* (1996), M/hPa | $\Delta H_r$ This Work, kJ/mol | $\Delta H_r$ Johnson *et al.* (1996), kJ/mol |
|---|---|---|---|---|---|
| Formic acid | 288 | 13.9 | 17.9 | -65 | -51 |
| | 298 | 5.6 | 8.8 | | |
| Acetic acid | 288 | 7.8 | 8.4 | -33 | -52 |
| | 298 | 4.9 | 4.1 | | |






| | | $K_H$ This Work, M/hPa | $K_H$ Johnson *et al.* (1996), M/hPa | $\Delta H_r$ This Work, kJ/mol | $\Delta H_r$ Johnson *et al.* (1996), kJ/mol |
|---|---|---|---|---|---|

**Table 5: Glycolaldehyde and acetic acid PCIMS reaction cell sensitivities (cps/ppb) for the 1700-7500 ppm reaction cell water vapor mixing ratio range. Glycolaldehyde sensitivities at m/z 92 ($O_2^-(GA)$) and m/z 187 ($I^-(GA)$) are for the Henry's Law source experiment, T = 288 K. Acetic acid microfluidic sensitivity at m/z 92 ($O_2^-(HAc)$) and m/z 187 ($I^-(HAc)$) are based on laboratory and field data presented in Figures 3 and 4. All sensitivities are reported from low to high water.**

| | | Sensitivity at m/z 92 (cps/ppb) | Sensitivity at m/z 187 (cps/ppb) |
|---|---|---|---|
| Glycolaldehyde | Case 1 & Case 2 | $8\text{-}20 \times 10^3$ | $8\text{-}10 \times 10^2$ |
| | Case 3 | $10\text{-}30 \times 10^4$ | $10\text{-}20 \times 10^3$ |
| Acetic Acid | Figure 3 | N/A | $1.4 - 1 \times 10^3$ |
| | Figure 4 | $1.4 - 1.6 \times 10^4$ | $1.4 - 1 \times 10^3$ |

**Table 6: Glycolaldehyde sensitivities for the melt vapor pressure source experiment, cps/ppb**

| Temperature (K) | $O_2^-$(GA) at m/z 92 | $I^-$(GA) at m/z 187 |
| --- | --- | --- |
| 298 | $6 \times 10^4$ | $7 \times 10^3$ |
| 318 | $7 \times 10^4$ | $1 \times 10^4$ |
| 338 | N/A | $1 \times 10^4$ |
| nominal | $6.5 \times 10^4$ | $9 \times 10^3$ |





| Temperature (K) | $O_2^-$(GA) at m/z 92 | $I^-$(GA) at m/z 187 |
| --- | --- | --- |

(a)

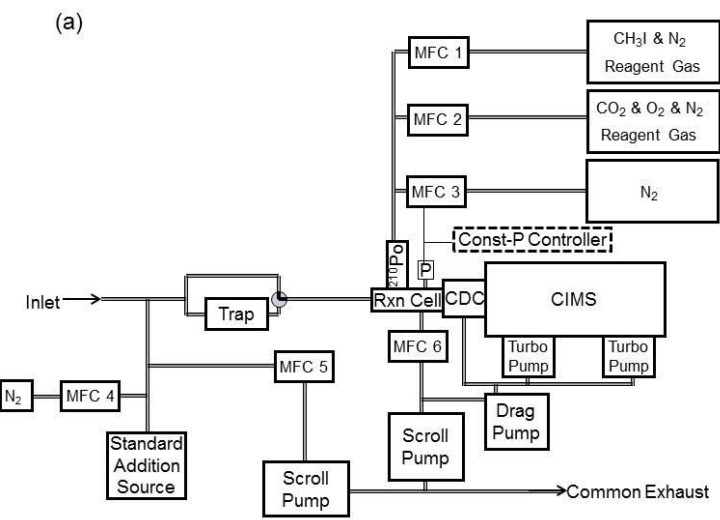

(b)

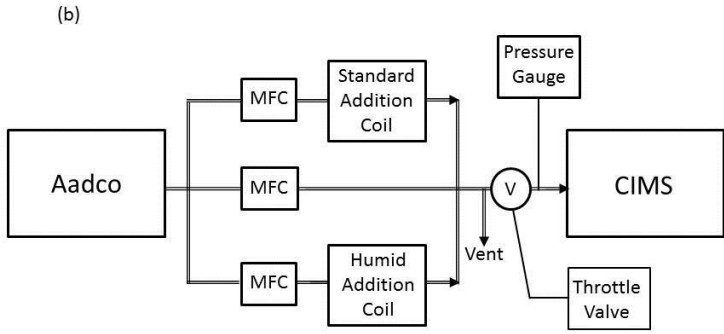


**Figure 1. (a) The peroxide chemical ionization mass spectrometer (PCIMS) instrument is diagramed in panel (a). The inlet samples either ambient air or laboratory generated pure air (Aadco Instruments Inc., Cleves, OH). "RXN" refers to the ion reaction cell. "CDC" refers to the octopole collision dissociation chamber and "MFC" indicates a mass flow controller and correspond to the**
**numbers in Table 1. "CIMS" represents the quadrupole mass spectrometer. (b) Laboratory calibration instrumental set up. "MFC" indicates a mass flow controller. "Aadco" is a pure air generator. "CIMS" in panel (b) represents the full PCIMS instrument illustrated in panel (a)**

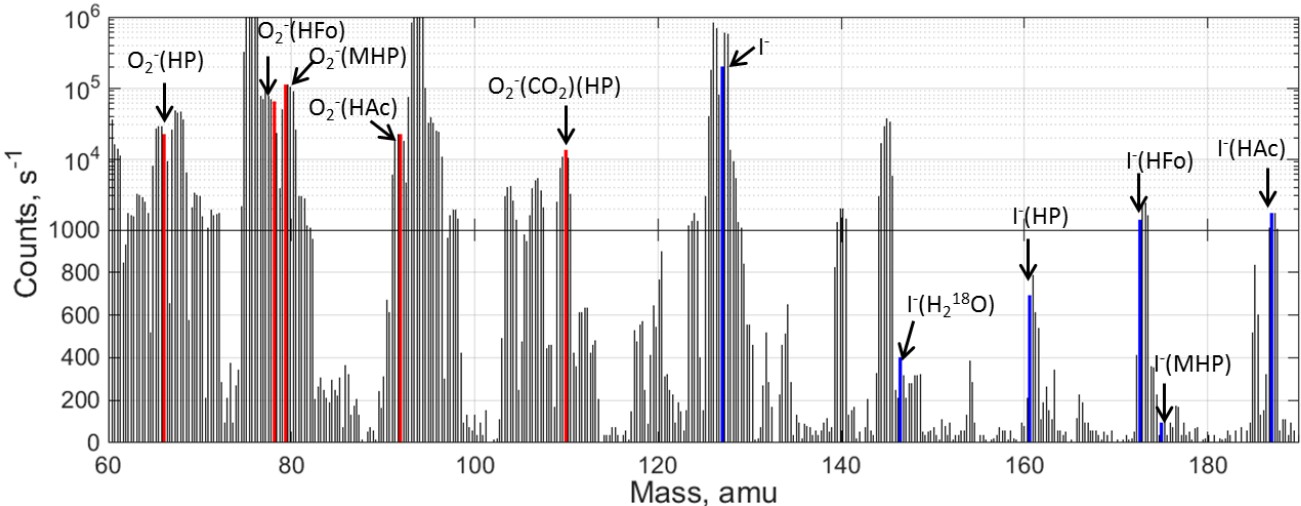

**Figure 2: PCIMS laboratory standard addition mass spectrum for the multi-reagent ion system showing the I⁻ and O₂⁻(CO₂) masses.** The PCIMS was operated at ambient pressure (1013 hPa) and a 370 ppm reaction cell water vapor mixing ratio. The mass dwell time was 50 milliseconds. The $O_2^-(CO_2)$ masses of interest are marked by red vertical lines and listed in increasing numerical order. These masses, and the corresponding ion clusters, are m/z 66 ($O_2^-(HP)$), m/z 78 ($O_2^-(HFo)$), m/z 80 ($O_2^-(MHP)$), m/z 92 ($O_2^-(HAc)$), and m/z 110 ($O_2^-(CO_2)(HP)$). The I⁻ masses of interest are marked by blue vertical lines and listed in increasing numerical order. These masses, and the corresponding ion clusters, are m/z 127 (I⁻), m/z 147 (I⁻($H_2^{18}O$)), m/z 161 (I⁻(HP)), m/z 173 (I⁻(HFo)), m/z 175 (I⁻(MHP)), and m/z 187 (I⁻(HAc)). Note the counts scale is linear up to 1000 and logarithmic above 1000.

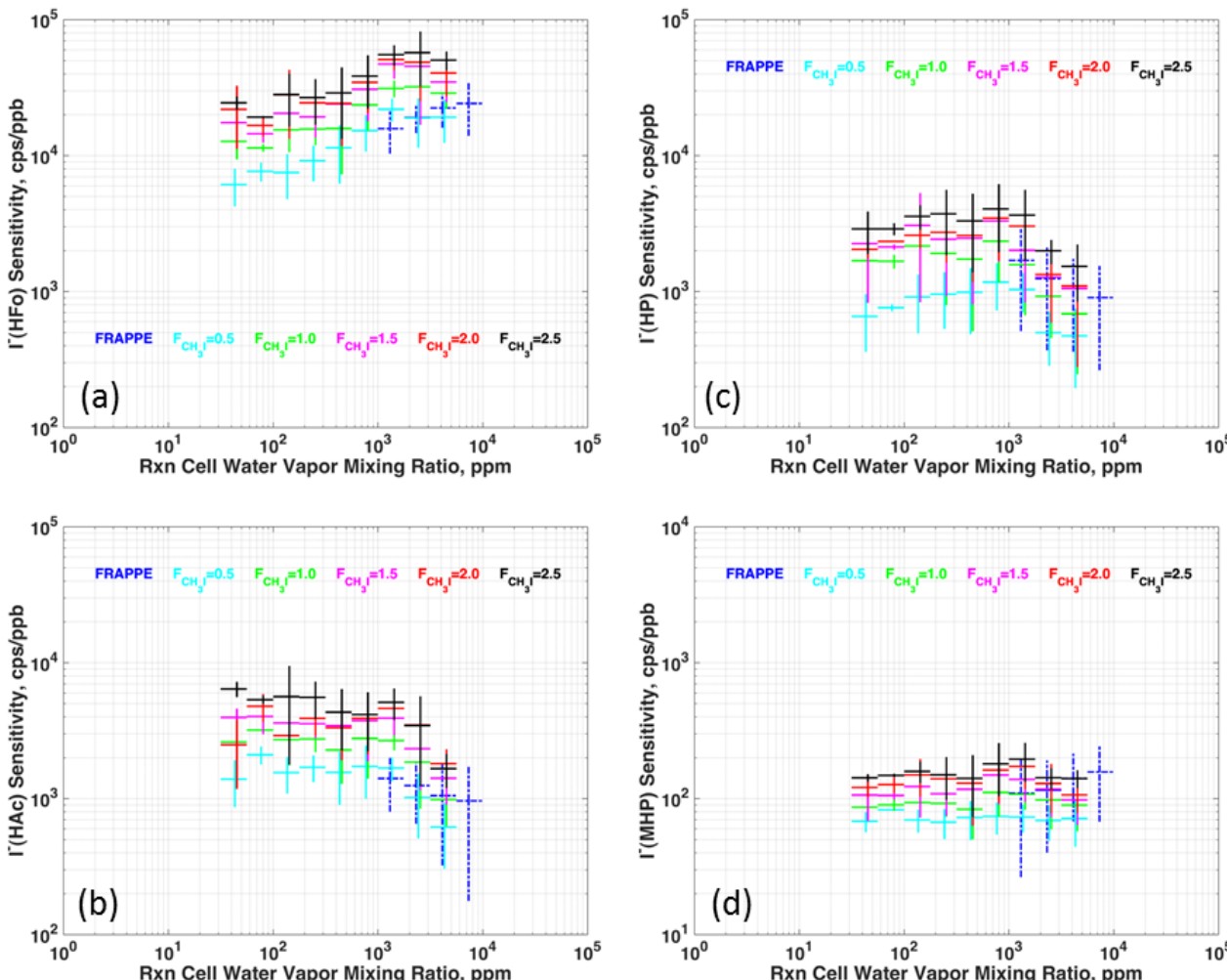

**Figure 3: Laboratory calibration sensitivities (cps/ppb) for five CH₃I flow rates (0.5 – 2.0 sccm) and FRAPPÉ in-flight calibration sensitivities as a function of reaction cell water vapor mixing ratio (ppm) for a) Γ(HFo) at m/z 173 b) ΓHAc at m/z 187, c) Γ(HP) at m/z 161, and d) Γ(MHP) at m/z 175. Note the scale difference for d. The horizontal bar represents the limits of the reaction cell water vapor mixing ratio bin and the mean sensitivity of that bin is plotted. The length of the vertical bar represents one standard deviation and the variability represents random variations in pressure, ambient concentrations during the standard addition, and systematic variations due to water vapor in a bin, calibration gas precision, and instrumental precision.**

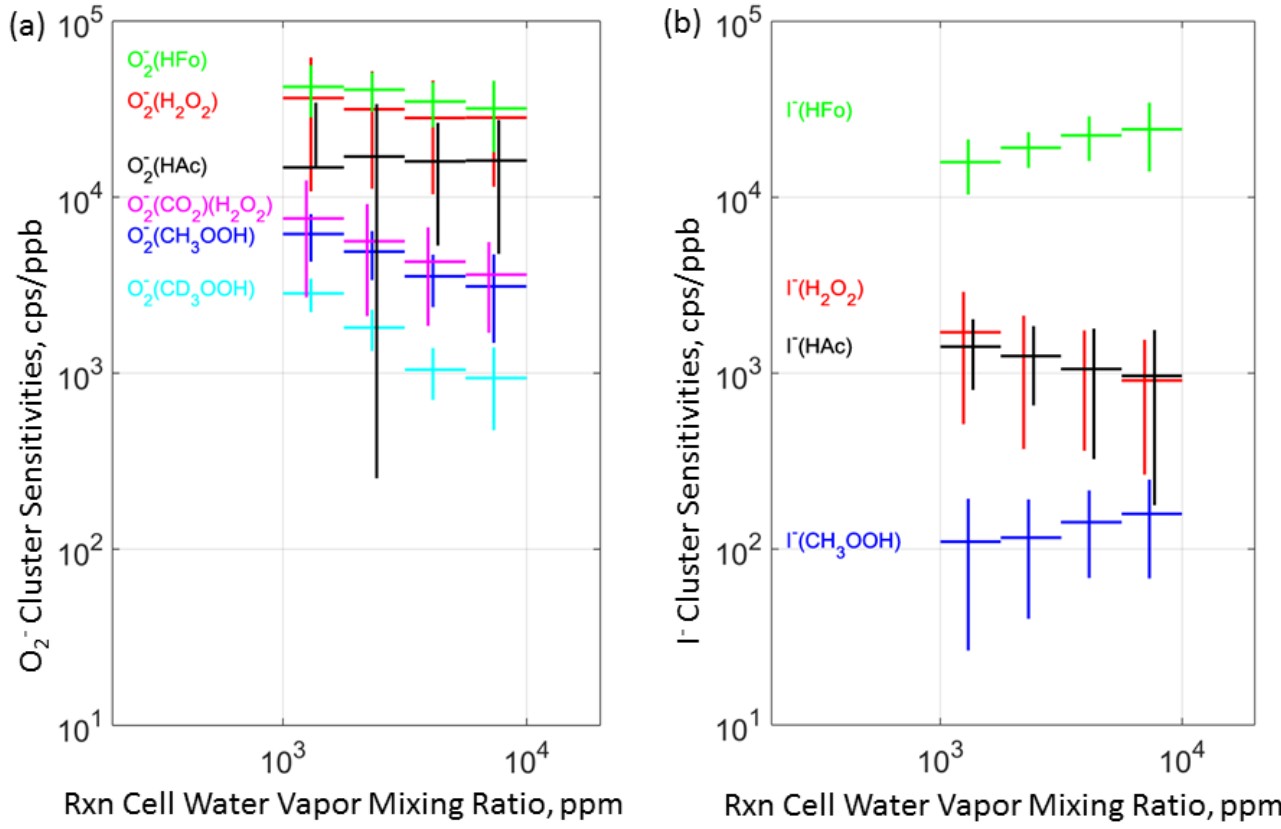

**Figure 4: FRAPPÉ in-flight sensitivities (cps/ppb) as a function of reaction cell water vapor for all PCIMS clusters. The left panel contains all the O$_2^-$ cluster and the right panel contains all the I$^-$ clusters. The horizontal and vertical error bars represent the same information as in Figure 3. O$_2^-$(CD$_3$OOH) refers to the deuterated MHP standard.**

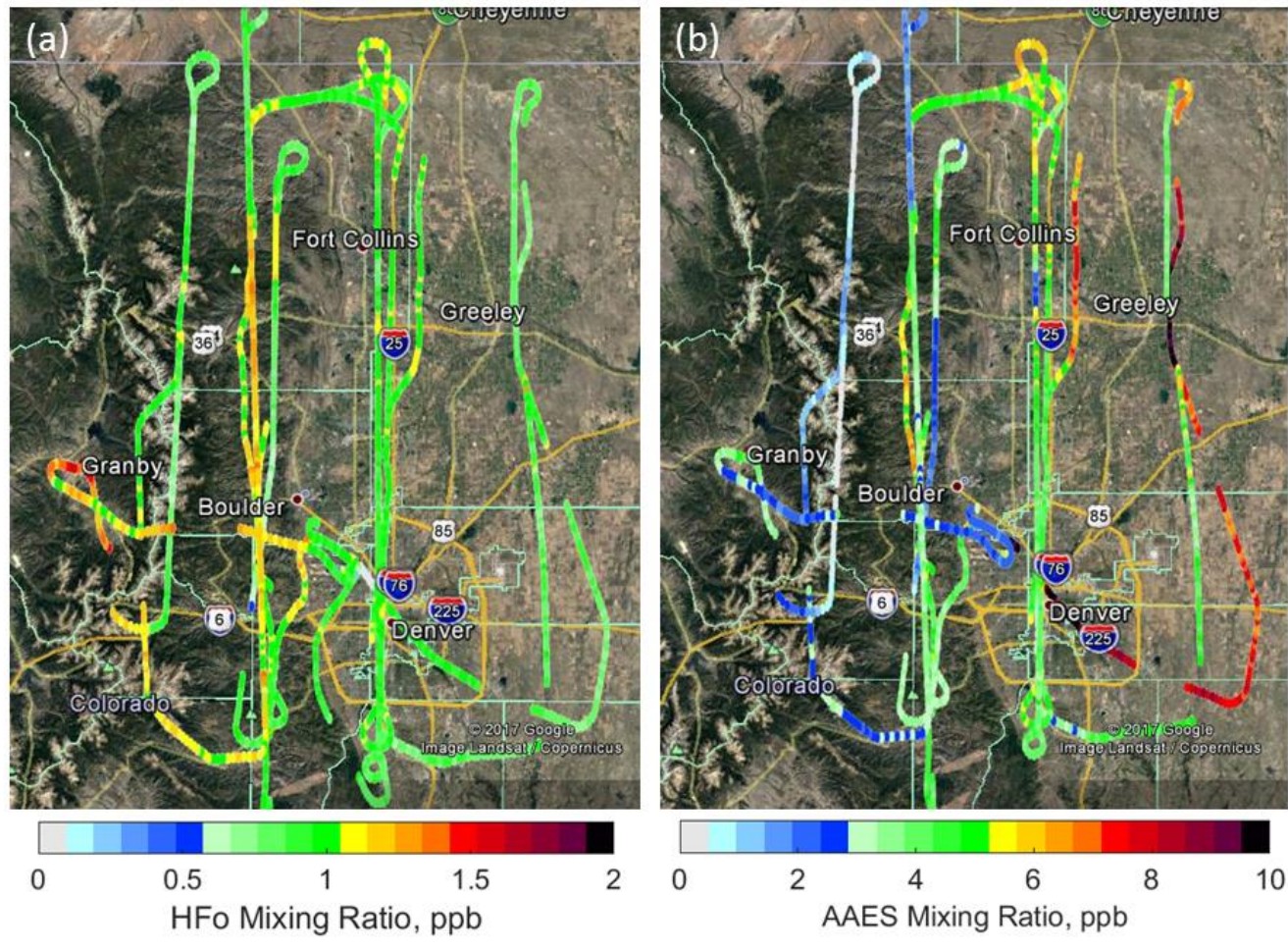

**Figure 5: Mixing ratios, as parts per billion (ppb), for a) formic acid (HFo), b) AAES (the sum of acetic acid and glycolaldehyde) for FRAPPÉ Research Flight 12 on August 12, 2014.**

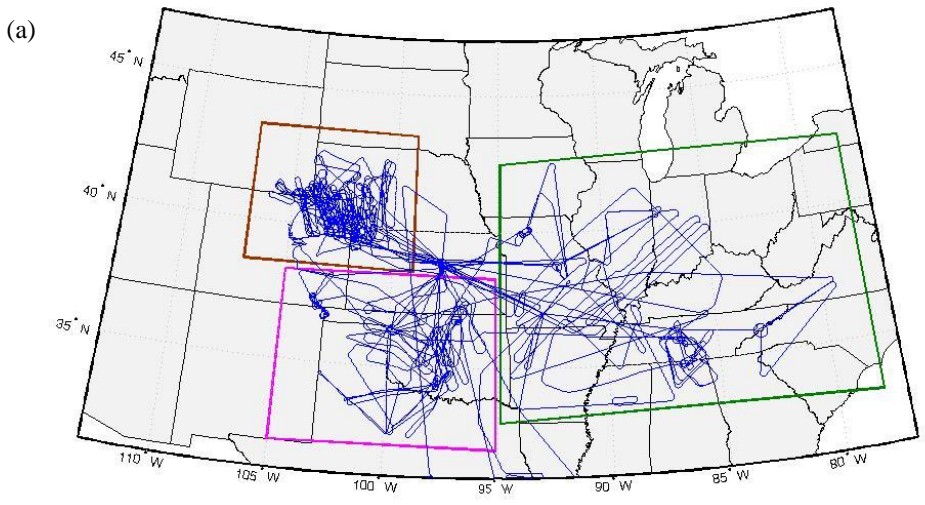

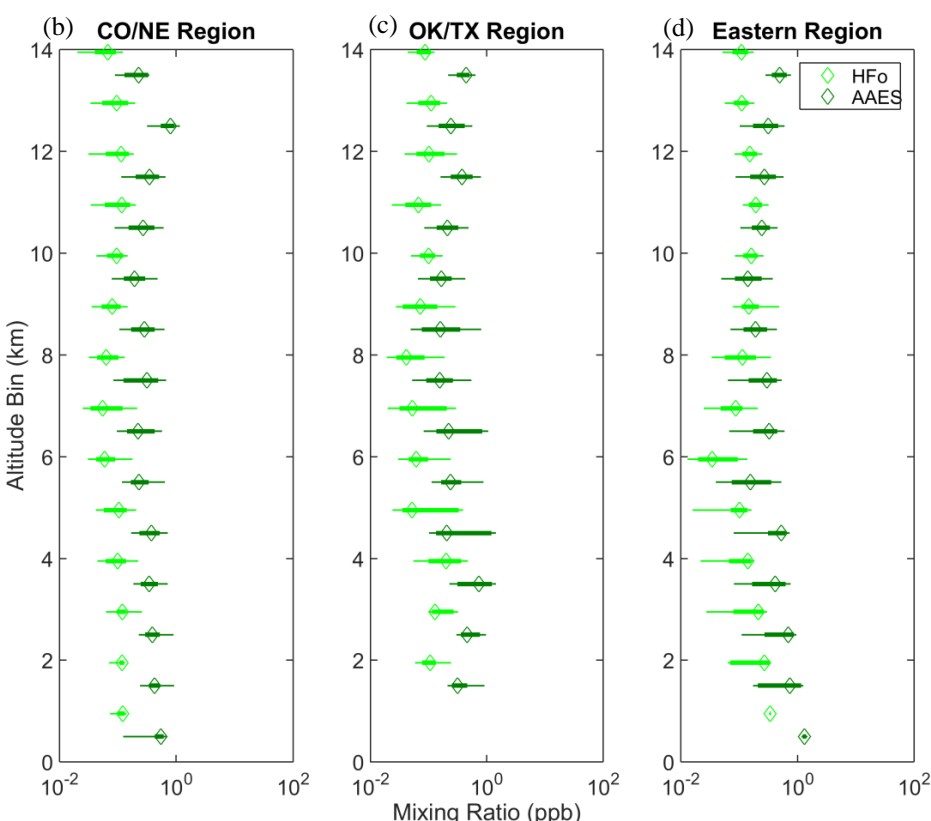

**Figure 6:(a) Map of three DC3 flight domains: Colorado-Nebraska (red), Oklahoma-Texas (magenta), and Eastern region (green) along with the HIAPER flight tracks, (b) Profiles for the HFo and AAES mixing ratios as a function of altitude for the three DC3 study regions (Colorado-Nebraska (CO/NE), Oklahoma-Texas (OK/TX), and Eastern Region). The symbols represent the median value for each altitude bin, the thick lines the interquartiles, and the thin line is the 10th – 90th percentile.**

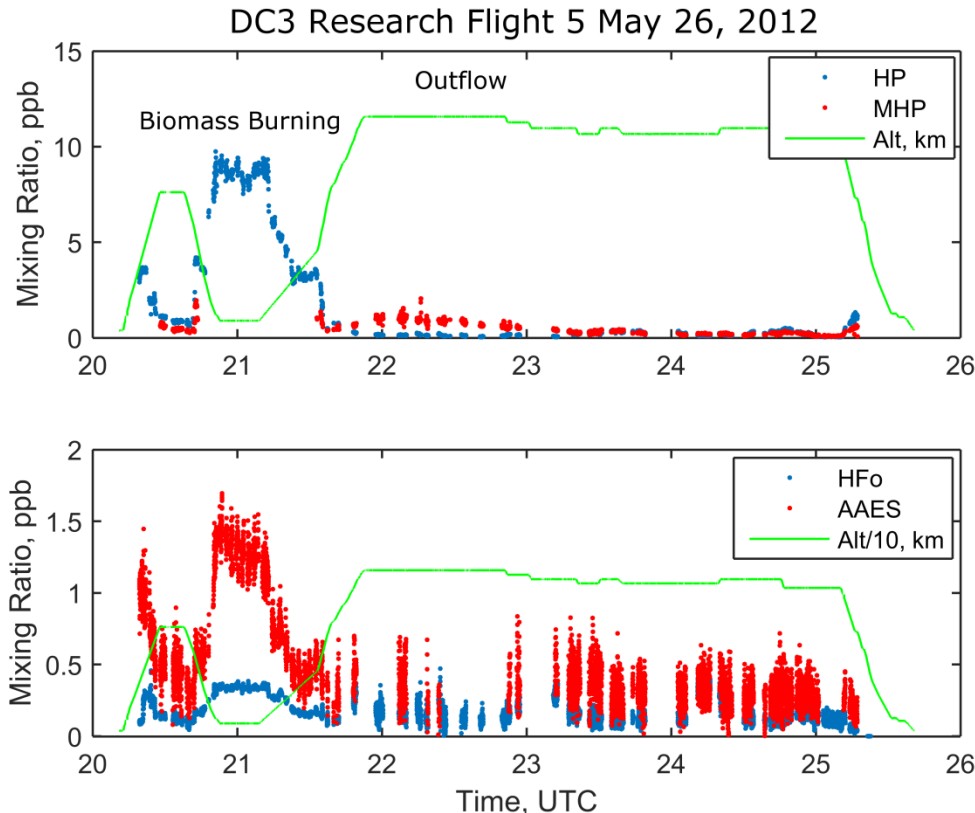

**Figure 7: PCIMS DC3 Research Flight 5 (May 26, 2012) sampling aged outflow from a Texas/Oklahoma storm. (top) Mixing ratios of HP (blue) and MHP (red) are shown in ppb as a function of flight time. (bottom) Mixing ratios of HFo (blue) and AAES (red) are shown in ppb as a function of flight time. The HIAPER altitude (green line) is in km (km/10 for the bottom figure). The periods of biomass burning and outflow are indicated. MHP is not reported during the low altitude leg due to potential interferences at mass 80 from $CO_3^-(H_2O)$ with an $^{18}O$ and $NO_3^-(H_2O)$.**