# Peer review of "Measurement of formic acid, acetic acid and hydroxyacetaldehyde, hydrogen peroxide, and methyl peroxide in air by chemical ionization mass spectrometry: airborne method development"

_Atmospheric Measurement Techniques, 2017_

## Referee Comment (RC1) · Anonymous Referee #1 · 4 Nov 2017

This study details the detection and quantification of a selection of important atmospheric molecules using a multi-reagent ion chemical ionization mass spectrometer (CIMS). The multi-reagent ion system reported here blends CO2 in air and CH3I in N2, with the primary reagent ions being O2-, CO2(O2)-, and I-. This is different from previously implemented multi-reagent ion systems, as the two reagent gases are added simultaneously and tuned such that I-, O2-, and CO2(O2)- ion cluster chemistries are operable. The multi-reagent system was successfully deployed in ambient air-borne and laboratory measurements. This novel twin-reagent CIMS technique is likely to be interesting to researchers working with CIMS instruments to detect important gas-phase atmospheric molecules. The less selectivity of O2- (and CO2(O2)-) CIMS coupled with the better understood I- CIMS has potential to improve the current understanding of atmospheric gas-phase chemistry. I recommend publication of the manuscript after some relatively minor issues (detailed below) are addressed.

ć Line 145: What is the reaction time between the sample gas and the reagent ions inside the ion-sample reaction cell?

aĂć Line 153: "This pressure was stated to provide the maximum yield of cluster ions and peak sensitivity..." Just checking, was this stated by the RXN cell manufacturer?

ć Line 165: Maybe mention the optimized mixing ratios of CO2 and pure air used for HP and MHP signals in addition to the reference to the O'Sullivan paper?

aĂć Figure 2: The figure presently does not provide a lot of information. What was the averaging time used to obtain the spectrum? Would a longer averaging time provide a less noisy spectrum with the relevant peaks clearly defined? I suppose the log scale was used to show the lower signals of I-(HP), I-(HFo), and I-(MHP) in the same figure. Maybe having a linear scale (so the highest peaks can be clearly shown), and a zoomed-in inset of these lower signals would make a better figure?

ć Trivial comment: Figure 4 has no a) and b) labels although it is referenced as such in the text.

åÅć Paragraph starting from line 399: Do you see an increase in signals of (H2O)nlclusters (where n is 2,3,4...) at above 1000 ppm water vapor mixing ratio? It could be that, at higher degrees of hydration of the I- anion, (HAc)I- formation becomes unfavorable (probably due to a steric hindrance to (HAc)I- formation, i.e. the multiple water molecules attached to I- make the formation of I-(HAc) difficult), causing a decrease in sensitivity. On the other hand, I-(HFo) formation might become more favorable when multiple water molecules are attached to I- (HFo being a smaller molecule might be additionally stabilized by a sequential evaporation of multiple water molecules), explaining the increase in I-(HFo) sensitivity at higher water vapor mixing ratios you report. In any case, the possible detection of water dimers, trimers, tetramers clustered to I- at high water vapor concentration should probably be commented upon.

åĂć Continuing on the same theme, I would think that the binding strength of (H2O)Icluster is weaker than the (HAc)I- cluster, so a ligand-exchange reaction between HAc and water, which is reaction 4 in your manuscript, is likely not the reason for the decrease in (HAc)I- signal at higher water vapor concentrations.

åĂć Line 402: "indicated the switching reaction equilibrium for HAc (4) behaved like that for HFo...". I might have misunderstood, but don't you observe an increase in the sensitivity of (HFo)I- with an increase in water vapor mixing ratio? Does that then not imply that (HFo)I-, unlike HAc, is not affected by a possible ligand exchange reaction with water (reaction 4)?

---

## Referee Comment (RC2) · Anonymous Referee #2 · 4 Dec 2017

**Review Treadaway et al AMTD 2017**

This paper describes calibrations, humidity tests, interference tests and initial results for the measurements using the PCIMS instrument on board the NSF/NCAR C130 aircraft during FRAPPE and the NSF/NCAR GV aircraft during DC3. The PCIMS is a chemical ionization quadrupole mass spectrometer using a mix of I- and O2- as primary ions to detect formic and acetic acid together with hydrogen peroxide and methyl peroxide. Currently the CIMS techniques are developing rapidly with the more commonly available high mass resolution ToF mass spectrometers together with easily adaptable

ion chemistries and papers such as this one are needed to explain the usefulness of the various versions of the CIMS instruments and used ion chemistries. The PCIMS still uses a quadrupole with unit mass resolution and suffers more from interferences and lower sensitivities than some of the current instruments. Nevertheless, I think this paper could be useful to the community, but clearly needs some improvements in organization and in presentation quality.

**Major/broader comments:**

- Why use multiple ion chemistries? It is not clear until quite late in the paper why a mixture of primary ions is used in this work and not only I-. Especially the abstract says: the focus is on organic acids and hydrogen peroxide and methyl peroxide react with I- as well. Later looking at Figure 3, one would pick O2- as the primary ion, because it has high sensitivity for all the compounds investigated here. Only on page 11 and 12 it becomes clear that O2- suffers from interferences and that methyl peroxide sensitivity is very low using I- only. This discussion needs to be moved to early in the paper to motivate the complicated ion chemistry setup.

- Why combined focus on acids and peroxides? In the introduction it should also be discussed why the PCIMS is focused on measuring the small acids and the small peroxides at the same time. Looking at recent literature, especially using I- chemistry a very large number of compounds can be measured. So why not focus on those and use a setup that improves the sensitivity and reduces the humidity dependence and generally simplifies the ion chemistry? Hydrogen peroxide seems to be detectable at sufficient sensitivity, only methyl peroxide will lack in sensitivity. So please give the reasoning for the focus on methyl peroxide, even though it significantly complicates the used ion chemistry.

- What is different in this paper than Heikes and O'Sullivan 2017 papers? There seems to be significant overlap with the Heikes et al and O'Sullivan et al 2017 (although not available yet) papers. Explain in detail what is different and new in this paper compared

to the two previous ones.

- The ambient data need other results to strengthen the arguments. During field campaigns like DC3 or FRAPPE lots of additional data are available. These data should be used here to strengthen the arguments made about the vertical profiles, identifying the biomass burning plumes, and biogenic influence. For example, do other measurements show the same C-shaped altitude profiles as the small acids? What are the enhancements of CO in the plume that is shown.

- GA calibration The weakest part of the paper is the order of magnitude uncertainty in the glycolaldehyde calibration. Glycolaldehyde is in most atmospheric environments only a minor contributor to the sum compared to acetic acid with biomass burning plumes being the notable exception, but nevertheless the glycolaldehyde sensitivity should be determined more accurately in this manuscript and not to only within a factor of ten. It is understood that producing a stable and accurate calibration source for glycolaldehyde is difficult, but a liquid calibration unit that completely nebulizes the GA/water solution (for example the LCU from Ionicon) or a catalytic conversion of GA to CO2 and subsequent CO2 detection (Veres et al 2010) would likely deliver much more accurate results than presented here. If no better calibration can be achieved, I would suggest to change the discussion in the manuscript such that an upper limit of the interference for acetic acid is used. In addition, the chapter of the GA calibration should be moved to the chapter 2.4: Calibration.

Minor/detailed comments:

Abstract: line 19: Add the ions that are used for the detection of HP and MHP.

Abstract: line 20: Already add here, why you are looking at HP and MHP, even though the focus seems to be on the small organic acids.

Page 2 line 58: add the values of the Henry's law constants to the text here.

Page 3 line 95-100: Delete this sentence: it is not relevant to the readers how you

СЗ

found out about this interference. It was pretty well known from previous literature.

Page 4 line 139: Please explain what "wetted" surfaces means.

Page 4 line 142: What is a "span check"?

Page 4 line 143: Please give more details on the organic acids and peroxide traps.

Page 5 lines 157-163: Please show how much the sensitivity changed with the inlet pressure.

Chapter 2.4: The uncertainty in the calibration should be added here somewhere.

Page 6 line 203-215: Describe only the trap that you used for FRAPPE and not the ones that did not work.

Figure 2: Figure 2 needs some improvement. First of all, please give the conditions (RH, press, ...) that this mass spectrum was recorded with. I suggest showing the mass spectrum as a stick plot, where all the masses are color coded according to their respective ion chemistry (I- or O2-). Then also label all the individual peaks with their mass and chemical composition.

All Figures: please be consistent with cts/s/ppb or cps/ppb. Just use one or the other, but don't mix them throughout the manuscript. There was also a switch to cps/ppt somewhere in the manuscript.

Figure S1 should be combined with Figure 3 in the main text. The FRAPPE results in Figure 3 are very difficult to distinguish from the rest of the data, please use a different symbol. Add the name of the compound or the ion to the y-axis label of each panel and not only give the mass. The same changes are needed for Figure 4. In general, for Figure 3 you are discussing the variation of the sensitivity caused by the flow of CH3I, so why is the flow not on the x-axis instead of the humidity. It would much easier to follow the discussion.

Tables 5-7 are all very small and should be combined into one larger Table.

Page 8 line 299: How much CH3I was present and how different was it during DC3 and FRAPPE?

Page 9 line 325-335: If there is a wide range of Henry's law constants in the literature, why did you compare in detail to the Johnson et al values, which are picked because they are the closest to the current measurements. It would me more important to discuss why there is such a wide range in the literature and why you think yours should be used here.

Page 10 line 343: The reasons for the lower sensitivity of the alcohols compared to the acids are very different for the PCIMS and the PTR-MS. I- does not cluster efficiently with alcohols whereas proton transfer from H3O+ causes fragmentation for the alcohols. This should be mentioned here, if you want to compare the two techniques. At this point I would also add that with a high resolution ToF, these compounds can be distinguished.

Page 10 line 346-365: As mentioned above: it would be good to describe the interference as an upper limit here and show how bad the interference could potentially be. Only if you are much more confident with one of the calibration cases, you can describe the interference with that one. The same for the discussion on Page 12.

Page 13 line 453: Also add Yuan et al 2017.

Figure 5: The flight tracks should be on top of a proper map showing the potential sources of the small acids such as urban areas and feed lots, all of which are readily available from FRAPPE.

Figure 7: indicate why there is such a large data gap in the HP data in the biomass burning plume.?

---

## Author Comment (AC1) · 29 Dec 2017

**Response to anonymous referee #1**

The authors thank the reviewer for their supportive feedback and constructive comments. We have responded to each comment below. The original comment is italicized and any additions or modifications to the manuscript are highlighted in red.

*Line 145: What is the reaction time between the sample gas and the reagent ions inside the ion-sample reaction cell?*

Response: The following sentence (revised manuscript line 152) has been added to help clarify the reaction cell mixing time.

The total flow through the reaction cell was fixed at 4.68 slpm (standard liters per minute; T = 273.15 K and P = 1013.25 hPa) and the mean transit time through the reaction cell was 17.8 ms (Heikes et al., 2017).

*Line 153: "This pressure was stated to provide the maximum yield of cluster ions and peak sensitivity…" Just checking, was this stated by the RXN cell manufacturer?*

Response: Yes it was stated by the instrument manufacturer THS.

The text (original manuscript line 151; revised manuscript line 157) has been modified to say: The RXN cell sample inlet and outlet critical orifices were of fixed diameter and optimized by THS to have a reaction cell pressure of 22 hPa, given the vacuum pumps and reagent gas system employed.

*Line 165: Maybe mention the optimized mixing ratios of CO2 and pure air used for HP and MHP signals in addition to the reference to the O'Sullivan paper?*

Response: The mixing ratios of $CO_2$ and $O_2$ change as a function of pressure (Table 3) and it is the 0.080 slpm flowrate that was selected by O'Sullivan.

The text (original manuscript line 165; revised manuscript line 171) has been improved to:
 The $CO_2$ and air reagent gas flow rate was optimized for HP and MHP signal response (O'Sullivan et al., 2017).

*Figure 2: The figure presently does not provide a lot of information. What was the averaging time used to obtain the spectrum? Would a longer averaging time provide a less noisy spectrum with the relevant peaks clearly defined? I suppose the log scale was used to show the lower signals of I-(HP), I-(HFo), and I-(MHP) in the same figure. Maybe having a linear scale (so the highest peaks can be clearly shown), and a zoomed-in inset of these lower signals would make a better figure?*

Response: Thank you for the feedback and suggestions on how to improve Figure 2. The comments from both reviewers about Figure 2 were considered and a modified figure was

prepared. The figure now has a linear scale from 0 – 1000 counts and then logarithmic from 1000 to $10^6$.

The following are the added in line text (revised manuscript line 285) describing the figure and the updated figure caption.

For this scan, the dwell time at each mass was 50 milliseconds, the ambient pressure was 1013 hPa, and the reaction cell water vapor mixing ratio was 370 ppm.

Figure 2: PCIMS laboratory standard-addition mass spectrum for the multi-reagent ion system showing the $I^-$ and $O_2^-(CO_2)$ masses. For this scan, the dwell time at each mass was 50 milliseconds, the ambient pressure was 1013 hPa, and the reaction cell water vapor mixing ratio was 370 ppm. The $O_2^-(CO_2)$ masses of interest are marked by red vertical lines and listed in increasing numerical order. These masses, and the corresponding ion clusters, are m/z 66 ($O_2^-($ HP)), m/z 78 ($O_2^-(HFo)$), m/z 80 ($O_2^-(MHP)$), m/z 92 ($O_2^-(HAc)$), and m/z 110 ($O_2^-(CO_2)(HP)$). The $I^-$ masses of interest are marked by blue vertical lines and listed in increasing numerical order. These masses, and the corresponding ion clusters, are m/z 127 ($I^-$), m/z 147 ($I^-($ $H_2^{18}O$)), m/z 161 ($I^-(HP)$), m/z 173 ($I^-(HFo)$), m/z 175 ($I^-(MHP)$), and m/z 187 ($I^-(HAc)$). Note the counts scale is linear up to 1000 and logarithmic above 1000.

*Trivial comment: Figure 4 has no a) and b) labels although it is referenced as such in the text.*

Updated

*Paragraph starting from line 399: Do you see an increase in signals of (H2O)nI clusters (where n is 2,3,4…) at above 1000 ppm water vapor mixing ratio? It could be that, at higher degrees of hydration of the I- anion, (HAc)I- formation becomes unfavorable (probably due to a steric hindrance to (HAc)I- formation, i.e. the multiple water molecules attached to I- make the formation of I-(HAc) difficult), causing a decrease in sensitivity. On the other hand, I-(HFo) formation might become more favorable when multiple water molecules are attached to I- (HFo being a smaller molecule might be additionally stabilized by a sequential evaporation of multiple water molecules), explaining the increase in I-(HFo) sensitivity at higher water vapor mixing ratios you report. In any case, the possible detection of water dimers, trimers, tetramers clustered to I- at high water vapor concentration should probably be commented upon.*

Response: Thank you for the suggestion and insight. Heikes et al. found that our pressure and humidity trends seen in the laboratory and field work for HP, HFo, and HAc could not be replicated without the addition of $I^-(H_2O)_2$ chemistry; however, $I^-$ $(H_2O)_2$ was not strong enough to survive declustering in the collision dissociation chamber. We do not generally record masses for $I^-(H_2O)_n$ but we do have scans from FRAPPE and the laboratory. There does not appear to be a response for $I^-$ $(H_2O)_2$ and it is unlikely that an n greater than 2 would survive the declustering.

The following text (revised manuscript line 394) has been added:
Heikes et al. found that the pressure and humidity trends seen in our PCIMS laboratory and field work for HP, HFo, and HAc could not be replicated without the addition of $I^-$ $(H_2O)_2$ (3),

especially at the higher humidity values. However, $I^-(H_2O)_2$ was not present in mass scans in FRAPPÉ or the laboratory and we inferred $I^-(H_2O)_2$ binding was not strong enough to survive declustering in the collision dissociation chamber.

*Continuing on the same theme, I would think that the binding strength of (H2O)I cluster is weaker than the (HAc)I- cluster, so a ligand-exchange reaction between HAc and water, which is reaction 4 in your manuscript, is likely not the reason for the decrease in (HAc)I- signal at higher water vapor concentrations.*

Response: You are correct if the system was flooded with $I^-$. Because we only add a small amount of $I^-$ to the system competition between the $H_2O$ and HAc is more pronounced than if the system had excess $I^-$.

*Line 402: "indicated the switching reaction equilibrium for HAc (4) behaved like that for HFo…" I might have misunderstood, but don't you observe an increase in the sensitivity of (HFo)I- with an increase in water vapor mixing ratio? Does that then not imply that (HFo)I-, unlike HAc, is not affected by a possible ligand exchange reaction with water (reaction 4)?*

Response: We do observe an increase in sensitivity of $I^-(HFo)$ with the addition of water but based on the work of Lee et al. (2014) it was thought that we would see a decrease in sensitivity at the highest humidities. As discussed in the manuscript the difference in instrumental set-ups and amount of $CH_3I$ present likely led to the difference in response.

To make this clearer for the HAc discussion the sentence (original manuscript line 401; revised manuscript line 419) has been altered to read:

[revised manuscript text omitted]
. The $Cu/NaHCO_3$ trap scrubbed the organic acids and HP but generated a positive trap response at m/z 80; the identity of which was not determined. The Carulite 200® trap was added after the $Cu/NaHCO_3$ trap which successfully eliminated the positive trap response at m/z 80. While this trap configuration removed both peroxides and organic acids the time response of the blank process was ~90 seconds and was considered too long for in flight blanks. The NaOH (5%) trap only removed organic acids. Running the air sample through the Carulite 200® and then the NaOH trap removed both peroxides and organic acids with minor

230 ~~outgassing. The blank equilibration time was much shorter (30-45 seconds) than with the Carulite 200® and Cu/NaHCO$_3$ trap. The Na$_2$CO$_3$ trap was a mixture of sodium carbonate (5 g), glycerol (5 g), water (250 mL), and methanol (250 mL) based on the EPA SOP (EPA, 2009). This trap was tested with and without the NaOH trap. Unfortunately, there was an outgassing of compounds which interfered at the masses used to measure HAc and HP. We speculate contaminants were present in the glycerol or methanol used to prepare the trap or from in situ chemical reactions leading to HAc and/or HP production on the~~

[revised manuscript text omitted]

**Table S2̶1̶: Reaction cell mixing ratios for glycolaldehyde based on Betterton and Hoffmann (1988) and Kua et al. (2010) in parts per billion (ppb) for the three experimental cases at the five tested equilibration air flow rates.**

| Aadco flowrate (sccm) | Glycolaldehyde Reaction Cell Mixing Ratio (ppb) | | |
|---|---|---|---|
| | Case 1 | Case 2 | Case 3 |
| 100 | 3.3 | 3.2 | 0.20 |
| 200 | 6.5 | 6.3 | 0.39 |
| 300 | 9.6 | 9.3 | 0.58 |
| 400 | 13 | 13 | 0.76 |
| 500 | 16 | 15 | 0.94 |

**Table S3:2 shows tThe expected GA reaction cell mixing ratio at different melt temperatures using the data from Petitjean et al. (2010).**

| Temperature (K) | Glycolaldehyde Reaction Cell Mixing Ratio (ppb) |
|---|---|
| 298 | 1.8 |
| 318 | 8.4 |
| 338 | 39 |
| 358 | 180 |

64

[Figure]

65    Figure S1: Laboratory calibration sensitivities (cps/ppb) as a function of CH₃I flow rate (slpm) for a) I⁻(HFo) at m/z 173 b) I⁻HAc

66    at m/z 187, c) I⁻(HP) at m/z 161, and d) I⁻(MHP) at m/z 175. The error bars represent one standard deviation and the variability

67    represents variations in pressure, reaction cell water vapor, calibration gas precision, and instrumental precision. Note the change

68    in the y-axis scale for c and d.

69

70

71

72

73

74

75

76

[Figure]

Figure S2: Laboratory calibration sensitivity (cps/ppb) as a function of $CH_3I$ flow rate (slpm) for $O_2^-$(MHP) at m/z 80. The error bars are the same as in Fig. S1.

[Figure]

93

94     **Figure S3**:   Calibration sensitivities ( cps/ppb of $O_2^-$(MHP) at m/z 80  from DC3 and laboratory work for five
95     $CH_3I$ flow rates (0.5 – 2.0 sccm).   The horizontal bar represents the limits of the reaction cell water vapor mixing ratio bin and the
96     mean sensitivity of that bin is plotted. The length of the vertical bar represents one standard deviation and the variability represents
97     random variations in pressure, ambient concentrations during the standard addition, and systematic variations due to water vapor in a
98     bin, calibration gas precision, and instrumental precision.

99

100

101

102

103

104

105

106

107

108

109

110

111

112

113

114

115

116

---

## Author Comment (AC2) · 29 Dec 2017

**Response to anonymous referee #2**

The authors thank the reviewer for their constructive comments and feedback. We have responded to each comment below. The original comment is italicized and any additions or modifications to the manuscript are highlighted in red.

**Major/broader comments:**

Why use multiple ion chemistries? It is not clear until quite late in the paper why a mixture of primary ions is used in this work and not only I-. Especially the abstract says: the focus is on organic acids and hydrogen peroxide and methyl peroxide react with I as well. Later looking at Figure 3, one would pick O2- as the primary ion, because it has high sensitivity for all the compounds investigated here. Only on page 11 and 12 it becomes clear that O2- suffers from interferences and that methyl peroxide sensitivity is very low using I- only. This discussion needs to be moved to early in the paper to motivate the complicated ion chemistry setup.

Why combined focus on acids and peroxides? In the introduction it should also be discussed why the PCIMS is focused on measuring the small acids and the small peroxides at the same time. Looking at recent literature, especially using I- chemistry a very large number of compounds can be measured. So why not focus on those and use a setup that improves the sensitivity and reduces the humidity dependence and generally simplifies the ion chemistry? Hydrogen peroxide seems to be detectable at sufficient sensitivity, only methyl peroxide will lack in sensitivity. So please give the reasoning for the focus on methyl peroxide, even though it significantly complicates the used ion chemistry.

Response: The first two questions will be answered together as the authors believe the topic is linked.

The opportunity presented itself to investigate the sensitivity of HFo and HAc to multiple ions,  $\Gamma$  and  $O_2^-$ , in the course of developing the PCIMS. HP and MHP were the primary targets during the development of the PCIMS instrument and HFo and HAc have been added during the modification post DC3. MHP was a critical species during the PCIMS development and it was important that we had sufficient sensitivity for it. A number of reagent ions were tested with HP and MHP and it was determined that  $O_2^-(CO_2)$  and  $O_2^-$  were best for the two peroxides. While  $\Gamma$  was effective in generating a cluster ion with HP, it yielded insufficient sensitivity towards MHP. A motivating factor for this work is the addition of HFo and HAc measurements to the DC3 campaign dataset, measurements which were not part of the original plan.

The following has been added to the introduction to provide some clarity and insight into why the multiple reagent ions are used and why it was important to maintain sufficient MHP sensitivity.

(revised manuscript line 87)

In the course of developing the PCIMS instrument, the opportunity presented itself to investigate the sensitivity of HFo and HAc to multiple reagent ions, specifically  $\Gamma$  and  $O_2^-$ .

(revised manuscript line 94)

 $\Gamma$ , derived from CH3I, proved to provide sufficient sensitivity for HP but not for MHP which was a critical species for the PCIMS, especially for the identification of deep convective storms during DC3 (O'Sullivan et al., 2017).

**And to the abstract (revised manuscript line 18):**

The CIMS also produced and detected  $\Gamma$  clusters with hydrogen peroxide and methyl peroxide,  $\Gamma$  (H2O2) and  $\Gamma$ (CH3OOH), though the sensitivity was lower than with the O2-(CO2) and O2- ion clusters, respectively. For that reason, while the  $\Gamma$  peroxide clusters are presented, the focus is on the organic acids.

What is different in this paper than Heikes and O'Sullivan 2017 papers? There seems to be significant overlap with the Heikes et al and O'Sullivan et al 2017 (although not available yet) papers. Explain in detail what is different and new in this paper compared to the two previous ones.

Response: O'Sullivan et al. reported the  $O_2^{-}(CO_2)$  laboratory and field work for HP and MHP. They did not discuss the iodide cluster work in any detail. Heikes et al. presented an ion-neutral chemical kinetic model to simulate the negative-ion chemistry presented in this paper and O'Sullivan et al. The goal of Heikes et al. was to establish a "theoretical basis to understand ambient pressure..., water vapor, ozone and oxides of nitrogen" (Heikes et al. 2017) of the peroxides and organic acids.

The introduction has been modified as follows (original manuscript line 108; revised manuscript line 111):

Heikes et al. (2017) and O'Sullivan et al. (2017) presented the PCIMS methodology for HP and MHP using  $O_2^{-}(CO_2)$  and  $O_2^{-}$ , respectively. O'Sullivan et al. (2017) presented PCIMS measurements for HP and MHP using  $O_2^{-}(CO_2)$  and  $O_2^{-}$ , respectively. Heikes et al. (2017) presented an ion-neutral chemical kinetic model to simulate the ion chemistry presented here and in O'Sullivan et al..

The ambient data need other results to strengthen the arguments. During field campaigns like DC3 or FRAPPE lots of additional data are available. These data should be used here to strengthen the arguments made about the vertical profiles, identifying the biomass burning plumes, and biogenic influence. For example, do other measurements show the same C-shaped altitude profiles as the small acids? What are the enhancements of CO in the plume that is shown.

Response: The authors agree with this assessment. However, extensive characterization about the PCIMS data is outside the scope of this work. There will be subsequent publications (as part of Treadaway's dissertation) that will discuss the organic acids during DC3 and FRAPPE in more detail. There will be a storm observation and modeling study published for DC3 and an organic acid source characterization study published for FRAPPE. Before these studies can proceed the authors feel it is necessary to publish the methodology behind collecting the organic acids with the PCIMS.

The authors concur that adding the CO enhancements in the plume are needed for identifying the biomass burning plume. The following has been added to the discussion: (revised manuscript line 504)

Biomass burning was identified by a CO enhancement of 80 ppb and HCN enhancement of >200 ppt above background.

And for the storm outflow (original manuscript line 484; revised manuscript line 506): The storm outflow portion (identified by MHP>HP)...

GA calibration The weakest part of the paper is the order of magnitude uncertainty in the glycolaldehyde calibration. Glycolaldehyde is in most atmospheric environments only a minor contributor to the sum compared to acetic acid with biomass burning plumes being the notable exception, but nevertheless the glycolaldehyde sensitivity should be determined more accurately in this manuscript and not to only within a factor of ten. It is understood that producing a stable and accurate calibration source for glycolaldehyde is difficult, but a liquid calibration unit that completely nebulizes the GA/water solution (for example the LCU from Ionicon) or a catalytic conversion of GA to CO2 and subsequent CO2 detection (Veres et al 2010) would likely deliver much more accurate results than presented here. If no better calibration can be achieved, I would suggest to change the discussion in the manuscript such that an upper limit of the interference for acetic acid is used. In addition, the chapter of the GA calibration should be moved to the chapter 2.4: Calibration.

Response: The authors agree that the large uncertainty in the glycolaldehyde is an issue. However, we disagree with the assessment that glycolaldehyde is only a minor contributor compared to acetic acid. It is true that based on reported literature values that glycolaldehyde's contribution could be smaller but there is very limited field observational data of glycolaldehyde. The authors do not feel comfortable treating the glycolaldehyde interference as an upper limit uncertainty for acetic acid with such limited observational data reported. The authors have included an additional table in the supplemental information (Table S1) and the text given with the Page 10 line 346-365 comment (below) showing the known list of surface and aircraft observations of gaseous glycolaldehyde. There has been more work done for biomass burning during laboratory work that is not included (e.g. Yokelson et al. 1997).

We appreciate the suggestions for determining a glycolaldehyde calibration source. We have been in contact with the Veres group but it seems unlikely that a catalytic conversion of glycolaldehyde to CO2 is possible at our concentration ranges. In addition, for both methods suggested there are issues with understanding the potential impurities and breakdown from the dimer into the monomer, trimer, etc in the aqueous phase as found by Petitjean et al. (2010) and Kua et al. (2013), respectively. Finally, it is unclear that a nebulizer would quantitatively generate GA gas or both GA gas and nanoaerosol (containing monomer GA and GA-hydrate, and the suite of dimer and trimer GA compounds identified in concentrated solutions).

We have moved the GA and alcohol calibration work into chapter 2.4.

**Minor/detailed comments:**

Abstract: line 19: Add the ions that are used for the detection of HP and MHP.

Response: Abstract (original manuscript line 19; revised manuscript line 18) was modified to: The CIMS also produced and detected  $\Gamma$  clusters with hydrogen peroxide and methyl peroxide,  $\Gamma$  (H2O2) and  $\Gamma$ (CH3OOH), though the sensitivity was lower than with the O2-(CO2) and O2- ion clusters, respectively. For that reason, while the  $\Gamma$  peroxide clusters are presented the focus is on the organic acids. Although hydrogen peroxide and methyl peroxide also form cluster ions with  $\Gamma$ , the focus here is on the organic acids.

Abstract: line 20: Already add here, why you are looking at HP and MHP, even though the focus seems to be on the small organic acids.

See above for addition to the abstract.

Page 2 line 58: add the values of the Henry's law constants to the text here.

Added

Page 3 line 95-100: Delete this sentence: it is not relevant to the readers how you found out about this interference. It was pretty well known from previous literature.

Response: The original sentence (original manuscript line 98; revised manuscript line 102) was deleted and the text has been modified to the following. However post FRAPPÉ, GA, a potential isobaric interference, was confirmed for I- chemistry with a relative response of approximately 1:1 for HAc:GA.

Page 4 line 139: Please explain what "wetted" surfaces means.

Response: The authors apologize for the use of jargon. A wetted surface is any surface (such as tubing) that comes into contact with the gas or liquid sample stream (containing the chemical we are trying to measure.

The text (original manuscript line 138; revised manuscript line 142) has been modified to clarify this:

The HIMIL and gas transfer lines were heated to 313 K in DC3 and 343 K in FRAPPÉ to minimize artifacts caused by the adsorption/release of the target gases onto/from the "wetted" inlet surfaces.

Any other references to "wetted" have been removed as well.

Page 4 line 142: What is a "span check"?

Response: This refers to selecting set masses to sample instead of scanning through all the masses.

The sentence (original manuscript line 142; revised manuscript line 145) has been changed to: The PCIMS responded linearly to the analyte gases measured at a fixed sample pressure and water vapor mixing ratio and species sensitivity was determined using a single calibration gas mixing ratio for each analyte.

Page 4 line 143: Please give more details on the organic acids and peroxide traps.

Response: The text (original manuscript line 142; revised manuscript line 147) has been modified.

Analytical blanks (Sect. 2.5) were determined by passing the sample air stream, with or without calibration gas, through serial It was determined that the combination of the Carulite 200® and NaOH traps. was best for scrubbing peroxides and organic acids

Page 5 lines 157-163: Please show how much the sensitivity changed with the inlet pressure.

Response: The experimental set up did not allow for inlet pressure to be independently varied while keep the reaction cell water vapor mixing ratio constant. Inlet pressure alters the sample flow rate and the reagent N2 flow rate and as a consequence  $O_2$ ,  $CO_2$ , and water vapor mixing ratios. Hence, reagent ion sensitivity changes due to pressure alone could not be quantified. Field sensitivity determinations also involved a convolution of inlet pressure and water vapor effects.

**Chapter 2.4: The uncertainty in the calibration should be added here somewhere.**

Response: Information has been added about the error in the FRAPPE organic acid aqueous standard and laboratory error in the sensitives.

The following have been added to Chapter 2.4.:

(revised manuscript line 182)

HFo and HAc standards (HCOOH, > 95% and CH3COOH, 99.9%, respectively) were obtained from Sigma-Aldrich. The HP standard was obtained from Fisher-Scientific (H2O2, 30%), and the MHP standard was synthesized in-house. (Lee et al., 1995).

**(revised manuscript line 194)**

During FRAPPÉ the organic acid aqueous standards were verified by titration (Treadaway, 2015). The percent errors between the theoretical and titrated concentrations were 1.00% and 1.51% for HFo and HAc, respectively. The FRAPPÉ peroxide aqueous standards, which were also used in post-mission, were standardized by titration and or UV absorbance

**(revised line 202)**

The average error in laboratory sensitivity for HFo and HAc was 26% and 31% respectively. This accounts for error in the PCIMS signal response and error in instrumental sources (e.g. mass flow controllers).

Page 6 line 203-215: Describe only the trap that you used for FRAPPE and not the ones that did not work.

Response: The discussion about traps used has been altered to remove superfluous information about traps not used.

**The changes in chapter 2.5 are as follows:**

Unfortunately, at low organic acid concentrations, there can be a positive trap response due to outgassing from the Carulite-200®. During FRAPPÉ Therefore, three different traps were tested as organic acid blank substrates: Cu/NaHCO3, Na2CO3, and NaOH. The Cu/NaHCO3 trap scrubbed the organic acids and HP but generated a positive trap response at m/z 80; the identity of which was not determined. The Carulite 200® trap was added after the Cu/NaHCO3 trap which successfully eliminated the positive trap response at m/z 80. While this trap configuration removed both peroxides and organic acids the time response of the blank process was ~90 seconds and was considered too long for in-flight blanks. It was determined that the NaOH (5%) trap was effective at removing organic acids but not peroxides. Running the air sample through the Carulite 200® and then the NaOH trap removed both peroxides and organic acids with minor outgassing. The blank equilibration time was much shorter (30-45 seconds) than with any of the other combinations tested (Treadaway, 2015)." The blank equilibration time was much shorter (30-45 seconds) than with the Carulite 200® and Cu/NaHCO3 trap. The Na2CO3 trap was a mixture of sodium carbonate (5 g), glycerol (5 g), water (250 mL), and methanol (250 mL) based on the EPA SOP (EPA, 2009). This trap was tested with and without the NaOH trap. Unfortunately, there was an outgassing of compounds which interfered at the masses used to measure HAc and HP. We speculate contaminants were present in the glycerol or methanol used to prepare the trap or from in situ chemical reactions leading to HAc and/or HP production on the Na2CO3 trap. It was determined that the combination of the Carulite 200® and NaOH traps was best for scrubbing peroxides and organic acids.

Figure 2: Figure 2 needs some improvement. First of all, please give the conditions (RH, press, ...) that this mass spectrum was recorded with. I suggest showing the mass spectrum as a stick plot, where all the masses are color coded according to their respective ion chemistry (I- or O2-). Then also label all the individual peaks with their mass and chemical composition.

Response: Thank you for the feedback and suggestions for Figure 2. The comments from both reviewers about Figure 2 were considered and a modified figure was prepared. The figure now has a linear scale from 0 - 1000 counts and then logarithmic from 1000 to  $10^6$ . The following are the added in line text (revised manuscript line 285) describing the figure and the updated figure caption.

For this scan, the dwell time at each mass was 50 milliseconds, the ambient pressure was 1013 hPa, and the reaction cell water vapor mixing ratio was 370 ppm.

Figure 2: PCIMS laboratory standard-addition mass spectrum for the multi-reagent ion system showing the  $\Gamma$  and  $O_2^{-}(CO_2)$  masses. For this scan, the dwell time at each mass was 50 milliseconds, the ambient pressure was 1013 hPa, and the reaction cell water vapor mixing ratio was 370 ppm. The  $O_2^{-}(CO_2)$  masses of interest are marked by red vertical lines and listed in increasing numerical order. These masses, and the corresponding ion clusters, are m/z 66 ( $O_2^{-}(HP)$ ), m/z 78 ( $O_2^{-}(HFo)$ ), m/z 80 ( $O_2^{-}(MHP)$ ), m/z 92 ( $O_2^{-}(HAc)$ ), and m/z 110 ( $O_2^{-}(CO_2)(HP)$ ). The  $\Gamma$  masses of interest are marked by blue vertical lines and listed in increasing numerical order. These masses, and the corresponding ion clusters, are m/z 127 ( $\Gamma$ ), m/z 147 ( $\Gamma$ ( H218O)), m/z 161 ( $\Gamma$ (HP)), m/z 173 ( $\Gamma$ (HFo)), m/z 175 ( $\Gamma$ (MHP)), and m/z 187 ( $\Gamma$ (HAc)). Note the counts scale is linear up to 1000 and logarithmic above 1000.

All Figures: please be consistent with cts/s/ppb or cps/ppb. Just use one or the other, but don't mix them throughout the manuscript. There was also a switch to cps/ppt somewhere in the manuscript.

Response: Thank you for catching this. The figures and text have been corrected so that cps/ppb is used consistently.

Figure S1 should be combined with Figure 3 in the main text. The FRAPPE results in Figure 3 are very difficult to distinguish from the rest of the data, please use a different symbol. Add the name of the compound or the ion to the y-axis label of each panel and not only give the mass. The same changes are needed for Figure 4. In general, for Figure 3 you are discussing the variation of the sensitivity caused by the flow of CH31, so why is the flow not on the x-axis instead of the humidity. It would much easier to follow the discussion.

Thank you for the suggestions on how to improve these figures and impact of iodide flow rate.

The FRAPPE data in Figures 3 and 4 were modified to be dashed blue lines which help to distinguish them from the laboratory work. The clusters are written on the y axis for Figures 3 and 4. Following the above suggestions additional supplemental figures have been added. The additions of the  $\Gamma$  clusters (Figure S1) and O2-(MHP) (Figure S2) are given for the laboratory calibration sensitivities as a function of CH3I flow rate. However, since the focus of this paper is on the  $\Gamma$  clusters the original Figure S1 (now S3) has been kept in the supplemental section. The O2-(MHP) cluster is only discussed in the reference of understanding the appropriate CH3I flow rate.

To help alleviate the confusion in the results section the text has been modified to discuss the  $CH_3I$  flow rates and then the water dependencies.

The updated portion of the results section (original manuscript starting line 288; revised manuscript starting line 293) is as follows:

This blended reagent ion system hinges on a balance between the iodide and oxygen chemistry. In general, as the proportion of  $CH_3I$  increased the sensitivity of the  $CO_2$  and  $O_2$  clusters decreased with the impact on MHP being greater than that for HP. The PCIMS is not as sensitive for HAc as for HFo (Figs. S1, 3, 4) and a sufficient amount of  $CH_3I$  is needed to promote clustering. Therefore, finding a balance between the two reagent gases ultimately depends on a prioritization between MHP and HAc. For this reason, five  $CH_3I$  flow rates (0.0005, 0.001, 0.0015, 0.002, and 0.0025 slpm) were evaluated. Figure S1 shows  $\Gamma$  cluster laboratory sensitivities for  $\Gamma$ (HFo),  $\Gamma$ (HAc),  $\Gamma$ (HP), and  $\Gamma$ (MHP) as a function of  $CH_3I$  flow rate. Figure S2 shows the laboratory MHP sensitivity at m/z 80 ( $O_2^{-1}$ (MHP)) as a function of  $CH_3I$  flow rate. All

of the pressure and water work is combined together which accounts for the large variance shown (1 standard deviation). The ion clusters' water dependencies are discussed below. As the CH3I flow rate increased, the O2-(MHP) sensitivity decreased. As expected, the sensitivities of the  $\Gamma$ (HFo),  $\Gamma$ (HAc),  $\Gamma$ (HP), and  $\Gamma$ (MHP) clusters increased as the CH3I flow rate increased with an approximate doubling in sensitivity for HFo and HP corresponding with a doubling in CH3I flow rate. Overall an increase in CH3I, and consequently  $\Gamma$ , resulted in an increase in  $\Gamma$ (HAc) sensitivity but at the cost of decreasing the O2-(MHP) sensitivity. It was fortuitous that there was enough CH3I present during DC3 to promote organic acid clustering without impairing the O2-(MHP) sensitivity. The data of Fig. S3,  $\Gamma$ (HP) in Fig. 3, and those for O2-(HP), O2-(CO2)(HP) (not shown) were used to identify the CH3I flow rate of 0.0005 slpm as providing the best sensitivity matches to the DC3 calibration data for HP and MHP.

Figure S1 S3 shows the MHP calibrations at  $m/z 80 (O_2^{-}(MHP))$  from DC3 as a function of reaction cell water vapor mixing ratio. Laboratory derived MHP sensitivity at m/z 80 is also shown as a function of reaction cell water vapor mixing ratio for 5 different CH3I flow rates (discussed below). The data are binned by the reaction cell water vapor mixing ratio. The mean sensitivity for that bin is plotted and the horizontal bar represents the limits of the reaction cell water vapor mixing ratio. The length of the vertical bar from the mean represents one standard deviation and includes random errors associated with variations in pressure, ambient concentrations during the standard addition, and systematic variations due to water vapor in a bin, calibration gas precision, and instrumental precision. As the CH3I flow rate increased, the  $O_2^{-}(MHP)$  sensitivity decreased. The data of Fig. S1, I-(HP) in Fig. 3, and those for  $O_2^{-}(HP)$ ,  $\Theta_2^{-}(CO_2)(HP)$  (not shown) were used to identify the CH3I flow rate of 0.0005 slpm as providing the best sensitivity matches to the DC3 calibration data for HP and MHP. It was fortuitous that there was enough CH3I present during DC3 to promote organic acid clustering without impairing the O2 (MHP) sensitivity. In future experiments, finding a balance between the two reagent gases ultimately depends on a prioritization between MHP and HAc. For this reason, five CH3I flow rates (0.0005, 0.001, 0.0015, 0.002, and 0.0025 slpm) were evaluated. Figure 3 shows I- cluster sensitivities for  $\Gamma(HP)$ ,  $\Gamma(MHP)$ ,  $\Gamma(HFo)$ , and  $\Gamma(HAc)$  for the FRAPPÉ experiment and from the same CH3I laboratory work as in Fig. <del>\$1</del> \$3. The horizontal and vertical error bars represent the same information as in Fig. S1 S3. As expected, the sensitivities of the I (HFo), I (HP), and F(MHP) clusters increased as the CH3I flow rate increased with an approximate doubling in sensitivity for HFo and HP with a doubling in CH3I flow rate. Also as anticipated, the HAc sensitivity increased substantially from the lowest to the highest CH3I flow rate though not always sequentially. Overall an increase in CH3I, and consequently I-, resulted in an increase in  $\Gamma(HAc)$  sensitivity but at the cost of decreasing the O2 (MHP) sensitivity.

Tables 5-7 are all very small and should be combined into one larger Table.

Response: Tables 5 and 6 were combined. Table 7 remains separate since it changed as a function of temperature to avoid confusion.

*Page 8 line 299: How much CH3I was present and how different was it during DC3 and FRAPPE?*

Response: A 5 ppm CH3I reagent gas was used with a 0.0005 slpm flow rate during FRAPPE. While CH3I was tried as a reagent gas in preparation of DC3 it was ultimately decided that it was not the best choice for a reagent ion for HP and MHP. However, we were fortuitous enough to detect  $\Gamma$  in the system as CH3I bled off from the tube walls. We used laboratory calibrations to determine the amount of CH3I present during DC3 using Figures S3 (originally S1), and  $\Gamma$ (HP) in Figure 3. This same amount of CH3I (0.0005 slpm) also provided a good balance between the O2- and  $\Gamma$  clusters. Hopefully the addition of Figures S1 and S2 and restructuring of the results section helped to elucidate this.

Page 9 line 325-335: If there is a wide range of Henry's law constants in the literature, why did you compare in detail to the Johnson et al values, which are picked because they are the closest to the current measurements. It would me more important to discuss why there is such a wide range in the literature and why you think yours should be used here.

Response: Johnson et al. data most closely compared to the Henry's Law values determined by us and provided the best connection between the microfluidic and coil laboratory work for the PCIMS. Further, the Johnson et al. values are also the only measured values reported in Sanders (2015) that experimentally determined the Henry's Law constant at more than one temperature.

The following sentence (revised manuscript line 336) has been added to the section to elaborate on our choice for the comparison.

Of the measured values reported in Sander (2015), only Johnson et al. (1996) experimentally determined the Henry's Law constants at multiple temperatures.

Page 10 line 343: The reasons for the lower sensitivity of the alcohols compared to the acids are very different for the PCIMS and the PTR-MS. I- does not cluster efficiently with alcohols whereas proton transfer from H3O+ causes fragmentation for the alcohols. This should be mentioned here, if you want to compare the two techniques. At this point I would also add that with a high resolution ToF, these compounds can be distinguished.

Response: We agree that there should be a clarification that the methods are different and the following text (revised manuscript line 355) has been added.

It should be acknowledged that these two techniques are different and some of the masses detected by the PTR-MS were fragments of the alcohols. While a time-of-flight CIMS can distinguish the alcohols from the organic acids (Yuan et al., 2016), there is a paucity of quadruple  $\Gamma$  CIMS data available with which to compare our  $\Gamma$  CIMS alcohol interference work.

Page 10 line 346-365: As mentioned above: it would be good to describe the interference as an upper limit here and show how bad the interference could potentially be. Only if you are much more confident with one of the calibration cases, you can describe the interference with that one. The same for the discussion on Page 12.

Response: Thank you for the suggestion but, as discussed above, the authors feel more comfortable using HAc and GA together as opposed to an upper limit interference for HAc. The following have been added to the manuscript to help clarify the need to report as AAES and not an upper limit for HAc.

Introduction (revised manuscript line 57): Table S1 provides a summary of literature surface and aircraft measurements for GA in urban, biomass burning, biogenic, and mixed environments. Results (revised manuscript line 377): As GA atmospheric mixing ratios are non-negligible (Table S1), PCIMS data collected at m/z 187 are reported as the "acetic acid equivalent sum," or AAES, of HAc plus GA.

**Page 13 line 453: Also add Yuan et al 2017.**

Response: Added the reference and updated the text (original manuscript line 452; revised manuscript line 470).

If the signal at m/z 187 were primarily HAc, the HAc:NH3 ratio was 0.078 ppb/ppb which is within the range reported by Paulot et al. (2011) though larger than the enhancement ratio range of 0.02-0.04 ppb/ppb reported by Yuan et al. (2017).

Figure 5: The flight tracks should be on top of a proper map showing the potential sources of the small acids such as urban areas and feed lots, all of which are readily available from FRAPPE.

Response: Thank you for the suggestion. The figure was altered and the text was altered as well to remove references to city codes

Figure 7: indicate why there is such a large data gap in the HP data in the biomass burning plume?

Response: We know that there is the potential for interference at mass 80 (MHP(O2-)) due to interferences of  $CO_3^-(H_2O)$  with an 18O and  $NO_3^-(H_2O)$ . For quality control we monitor mass 78 for  $CO_3^-(H_2O)$  and mass 98 for  $NO_3^-(H_2O)_2$ . If either 78 or 98 are high the MHP data is not reported. This is discussed in full in both in Heikes et al. (2017) and O'Sullivan et al. (2017). The following text was added in the text and the figure caption for Figure 7 to clarify why there is a data gap.

In text (revised manuscript line 504): There is no MHP reported during this period due to potential interferences at mass 80 from  $CO_3^{-}(H_2O)$  with an 18O and/or  $NO_3^{-}(H_2O)$  (Heikes et al., 2017).

Figure 7: MHP is not reported during the low altitude leg due to potential interferences at mass 80 from  $CO_3^{-}(H_2O)$  with an 18O and/or  $NO_3^{-}(H_2O)$ .

**Measurement of formic acid, acetic acid and hydroxyacetaldehyde, hydrogen peroxide, and methyl peroxide in air by chemical ionization mass spectrometry: airborne method development**

5 Victoria Treadaway1, Brian G. Heikes1, Ashley S. McNeill1,2, Indira K.C. Silwal3,4 and Daniel W. O'Sullivan3

[revised manuscript text omitted]
É, it was brought to our attention that GA-was, a potential exact-isobaric interference-in the measurement of HAc (Wisthaler, pers. comm. 2015) and we needed to evaluate this possibility for our Γ-CIMS chemistry. As presented below, this potential interference was subsequently, was confirmed for Γ chemistry with a relative response of approximately 1:1 for HAc:GA.- We necessarily report the m/z 187 signal as the "acetic acid equivalent sum" (AAES) of HAc and GA in our prior DC3 and FRAPPÉ datasets (data reporting in progress).
- This study details the detection and quantification of HFo and AAES using a multi-reagent ion CIMS. The multi-reagent ion PCIMS is unique as it allows the detection of HFo and AAES, as well as, hydrogen peroxide (H2O2, hereafter referred to as HP) and methyl peroxide (CH3OOH, hereafter referred to as MHP). The multi-reagent ion gas system blends a CO2 in air mixture and a CH3I in N2 mixture with pure N2. This is different from other multi-reagent ion systems such as Brophy and Farmer (2015) as the two reagent gases are added simultaneously and tuned such that I-, O2-, and O2-(CO2) ion cluster chemistries are operable. HeikesO'Sullivan et al. (2017) and O'Sullivan et al. (2017) presented the PCIMS methodologymeasurements 
[revised manuscript text omitted]
 N2 gas (Scott-Marrin). This CH3I mixture was further diluted with N2 to a 5 ppm CH3I mixing ratio which was found to reproduce the field sensitivities of HP, MHP, and H218O observed in DC3 (Treadaway, 2015). The final reagent gas blend of CH3I, CO2, O2, and N2 yielded responses for Γ, O2-, and O2-(CO2) cluster ions with organic acids, peroxides, hydroxyacetaldehyde, and water vapor.

**2.4 Calibration Configuration**

HFo and HAc standards (HCOOH, > 95% and CH3COOH, 99.9%, respectively) were obtained from Sigma-Aldrich.
 The HP standard was obtained from Fisher-Scientific (H2O2, 30%), the MHP standard was synthesized and dilutions of both

were standardized by titration and or UV absorbance (Lee et al., 1995a). In-flight calibrations were performed by microfluidic injection. Two versions of the microfluidic system were used to inject the liquid standard into the PCIMS via a nitrogen gas line. For the first set-up, used during DC3, the standard was contained in a Hamilton glass syringe and injected using a single syringe pump ( $1 \times 10^{-6}$  L/min aqueous flow rate, KD Scientific Inc., Holliston, MA). The liquid standard was vaporized in a heating

- 195 block (328 K) into a gaseous  $N_2$  stream (0.4 slpm). The disadvantage of this system is that it can only calibrate for peroxides or organic acids and was used exclusively for the peroxides, as they were the target analytes of interest. After DC3, a second microfluidic system was developed which allowed for calibration of peroxides and organic acids. Both peroxide and organic acid aqueous standards (in Hamilton glass syringes) were injected (5 x 10-7 L/min) and evaporated into a  $N_2$  gas stream (0.4 slpm) via mixing-Ts and a ballast PFA-Teflon® mixing vessel. Both microfluidic standard addition systems were contained
- 200 within the PCIMS instrument rack. In-flight calibrations were done as part of the FRAPPÉ program in the summer of 2014 with the second microfluidic set-up. -During FRAPPÉ the organic acid aqueous standards were verified by titration (Treadaway, 2015). The percent errors between the theoretical and titrated concentrations were 1.00% and 1.51% for HFo and HAc, respectively. The FRAPPÉ peroxide aqueous standards, which were also used in post-mission laboratory work, were standardized by titration and/or UV absorbance with an estimated accuracy of 5% and 10%, respectively.
- 205

210

Sensitivities were determined in-flight by standard addition. The ambient signal before and after the calibration gas addition was used to estimate the ambient signal at the time of calibration gas addition. The sensitivity was then determined by dividing the calibration gas mixing ratio in the reaction cell by the difference between the combined standard addition and ambient signal and the interpolated ambient signal. The sensitivity of each compound is reported as counts per second per ppb (cps/ppb). The average error in laboratory sensitivity for HFo and HAc was 26% and 31% respectively. This accounts for error in the PCIMS signal response and error in instrumental sources (e.g. mass flow controllers).

Henry's Law constants were determined for HFo and HAc using a gas-aqueous coil equilibrium apparatus. HFo (0.3 mM) and HAc (0.9 mM), were acidified (0.02 N  $H_2SO_4$ ) to keep each acid in its protonated form and thereby ensure partitioning into the gas phase according to each acid's Henry's Law constant. Henry's Law constants from Johnson et al. (1996)Johnson et al. (1996) were used. Zero air (0.2 or 0.4 slpm) was passed through an equilibration coil in a water bath kept at 288 or 298 K

along with the organic acid standard. The resulting calibration gas was added to the sample air stream after humidification (Sect. 2.6). For the work at 298 K, the laboratory room temperature was increased to 303 K to prevent water vapor from condensing on the transfer tubing walls. This same set-up was used for the GA Henry's Law experiment and the alcohol interference work described below.

**2.5 Blank Configuration**

Carulite-200® (Carus Corporation, Peru, IL), a magnesium dioxide/copper oxide catalyst, is an effective ozone and peroxide destruction catalyst and was used during DC3 as an analytical blank substrate for the peroxides (O'Sullivan et al., 2017). It further proved to be effective in removing but not destroying the organic acids as well. Unfortunately, at low organic acid concentrations, there can be a positive trap response due to outgassing from the Carulite-200®.

225

220

During FRAPPÉ three different traps were tested as organic acid blank substrates: Cu/NaHCO3, Na2CO3, and NaOH. The Cu/NaHCO3 trap scrubbed the organic acids and HP but generated a positive trap response at m/z 80; the identity of which was not determined. The Carulite 200® trap was added after the Cu/NaHCO3 trap which successfully eliminated the positive trap response at m/z 80. While this trap configuration removed both peroxides and organic acids the time response of the blank process was ~90 seconds and was considered too long for in flight blanks. The NaOH (5%) trap only removed organic acids. Running the air sample through the Carulite 200® and then the NaOH trap removed both peroxides and organic acids with minor

- 230 outgassing.-The blank equilibration time was much shorter (30 45 seconds) than with the Carulite 200® and Cu/NaHCO3 trap. The Na2CO3 trap was a mixture of sodium carbonate (5 g), glycerol (5 g), water (250 mL), and methanol (250 mL) based on the EPA SOP (EPA, 2009). This trap was tested with and without the NaOH trap. Unfortunately, there was an outgassing of compounds which interfered at the masses used to measure HAc and HP. We speculate contaminants were present in the glycerol or methanol used to prepare the trap or from in situ chemical reactions leading to HAc and/or HP production on the
- 235

245

Na2CO3 trap. It was determined that the combination of the Carulite 200® and NaOH traps was best for scrubbing peroxides and organic acids.

[revised manuscript text omitted]
. As the CH2I flowrate increased, the O2-(MHP) sensitivity decreased. The data of Fig. S1, I-(HP) in Fig. 3, and those for O2-(HP), O2-(CO2)(HP) (not shown) were used to identify the CH2I flow rate of 0.0005 slpm as providing the best sensitivity matches to the DC3 calibration data for HP and MHP. It was fortuitous that there was enough CH2I present during DC3 to promote organic acid clustering without impairing the O2-(MHP) sensitivity.
- In future experiments, finding a balance between the two reagent gases ultimately depends on a prioritization between MHP and HAc. For this reason, five CH3I flowrates (0.0005, 0.001, 0.0015, 0.002, and 0.0025 slpm) were evaluated. Figure 3 shows I cluster sensitivities for I'(HP), I'(MHP), I'(HFo), and I'(HAc) for the FRAPPÉ experiment and from the same CH3I laboratory work as in Fig. S1. The horizontal and vertical error bars represent the same information as in Fig. S1. As expected, the sensitivities of the I'(HFo), I'(HP), and I'(MHP) clusters increased as the CH3I flowrate increased with an approximate doubling in sensitivity for HFo and HP with a doubling in CH3I flowrate. Also as anticipated, the HAc sensitivity increased substantially from the lowest to the highest CH3I flowrate though not always sequentially. 
[revised manuscript text omitted]

3CO2 (400 ppm) in ultrapure air (Scott-Marrin).

4CH3I (5 ppm) in ultrahigh purity N2 (Scott-Marrin)

| Table 2: Laboratory detection limits (ppt) | determined as three times the standard | d deviation of the blank using | g a pure air system as a |
|--------------------------------------------|----------------------------------------|--------------------------------|--------------------------|
| function of sample inlet pressure (hPa)    |                                        |                                |                          |

| Pressure, hPa | HFo, ppt | HAc, ppt |
|---------------|----------|----------|
| 120           | 46       | 86       |
| 180           | 23       | 46       |
| 306           | 13       | 37       |
| 600           | 18       | 59       |
| 1013          | 59       | 120      |

**Table 3: Laboratory instrument calibration conditions: sample inlet pressure, reaction cell water vapor mixing ratios, and reagent gas1005reaction cell mixing ratios**

| Sample
Pressure, hPa | Reaction Cell Water Vapor
Mixing Ratio 1 , ppm |                   | Reaction Cell Reagent Gas Mixing Ratio |                       |                      | atio                 |
|-------------------------|--------------------------------------------------------------|-------------------|----------------------------------------|-----------------------|----------------------|----------------------|
|                         | Low                                                          | High              | CH 3 I, ppb                 | CO 2 , ppm | O 2 , ppm | N 2 , ppm |
| 120                     | 40                                                           | 540               | 0.575                                  | 7.36                  | 3678                 | 996322               |
| 180                     | 50                                                           | 610               | 0.580                                  | 7.42                  | 3712                 | 996288               |
| 306                     | 90                                                           | 1100              | 0.616                                  | 7.88                  | 3941                 | 996059               |
| 600                     | 230                                                          | $4400^{2}$        | 0.814                                  | 10.42                 | 5212                 | 994788               |
| 1013                    | 370                                                          | 7700 2 | 1.174                                  | 15.02                 | 7512                 | 992488               |

1This work was performed with a water bath at 288 K

 $^2 {\rm This}$  includes work in a water bath at 298 K

Table 4: Henry Law constants and enthalpies for formic and acetic acid

| Species     | Temperature,
° K | K H This Work,
M/hPa | K H Johnson et
al. (1996),
M/hPa | ΔH r, This Work,
kJ/mol | ΔH r , Johnson et
al . (1996),
kJ/mol |
|-------------|----------------------------|------------------------------------|-----------------------------------------------------------------|---------------------------------------|----------------------------------------------------------------------|
| Formic acid | 288                        | 13.9                               | 17.9                                                            | -65                                   | -51                                                                  |
|             | 298                        | 5.6                                | 8.8                                                             | -05                                   | -51                                                                  |
| Acetic acid | 288                        | 7.8                                | 8.4                                                             | -33                                   | -52                                                                  |
|             | 298                        | 4.9                                | 4.1                                                             |                                       | 52                                                                   |

Table 5: Henry's Law source experiment glycolaldehydeGlycolaldehyde and acetic acid PCIMS reaction cell sensitivities at T=288K, ets/s/ppb (cps/ppb) for the 1700-7500 ppm reaction cell water vapor mixing ratio range. Glycolaldehyde sensitivities at m/z 92 (Q2 1060 (GA)) and m/z 187 (F(GA)) are for the Henry's Law source experiment, T = 288 K. Acetic acid microfluidic sensitivity at m/z 92 (O2) (HAc)) and m/z 187 (F(HAc)) are based on laboratory and field data presented in Figures 3 and 4. All sensitivities are reported from low to high water. O2 (GA)Sensitivity at m/z 92 I-(GA)Sensitivity at m/z 187 (cps/ppb) (cps/ppb) Case 1 & Case 2 <del>8-20 x 103</del>  $8.10 \times 10^{2}$ Sensitivity, (cps/ppb)Glycolaldehyde  $10 - 30 \times 10^4$  $10.20 \times 10^3$ Case 3 1065 <del>N/A</del>  $\frac{-1 \times 10^3}{3}$ 1.4 Figure 3 Acetic Acid  $1.4 - 1.6 \times 10^4$ 1.4 $\frac{1 \times 10^3}{1 \times 10^3}$ Figure 4 Table 5: Glycolaldehyde and acetic acid PCIMS reaction cell sensitivities (cps/ppb) for the 1700-7500 ppm reaction cell water vapor mixing ratio range. Glycolaldehyde sensitivities at m/z 92 (O2 (GA)) and m/z 187 (I (GA)) are for the Henry's Law source experiment, 1070 T = 288 K. Acetic acid microfluidic sensitivity at m/z 92 ( $\overline{O_2}$  (HAc)) and m/z 187 ( $\Gamma$  (HAc)) are based on laboratory and field data presented in Figures 3 and 4. All sensitivities are reported from low to high water. Sensitivity at m/z 92 (cps/ppb) Sensitivity at m/z 187 (cps/ppb)  $8-20 \times 10^3$ 8-10 x 102 Case 1 & Case 2 Glycolaldehyde 10-30 x 104 Case 3  $10-20 \ge 10^3$ N/A  $1.4 - 1 \ge 10^3$ Figure 3 Acetic Acid  $1.4 - 1.6 \ge 10^4$  $1.4 - 1 \ge 10^3$ Figure 4 1075

1080

| Temperature (K) | O 2 -(GA) at m/z 92 | I'(GA) at m/z 187   |
|-----------------|--------------------------------|---------------------|
| 298             | 6 x 10 4            | 7 x 10 3 |
| 318             | 7 x 10 4            | 1 x 10 4 |
| 338             | N/A                            | 1 x 10 4 |
| nominal         | $6.5 \ge 10^4$                 | 9 x 10 3 |

 Table 6: Glycolaldehyde sensitivities for the melt vapor pressure source experiment, <a href="https://ets/scipation.org">ets/scipation.org</a>/ppb

---

## Referee Report (RR1)

Review of the revised paper: Treadaway et al

For this version of the paper the authors have taken all my comments into account. The organization of the paper and the figures are much improved. The uncertainty in the glycolaldehyde is the weakest part of the paper, but I don't think this can be improved in this paper. A good calibration for GA is a research project on its own. So I don't have any further comments and the paper can now be accepted as is.